# *Fix the Mind, Not the Move:*
# Interpretable AI Assistance via Knowledge-Gap Localization

**Ayano Hiranaka** [* 1]   **Ya-Chuan Hsu** [* 1]   **Stefanos Nikolaidis** [1]   **Erdem Bıyık** [† 1]   **Daniel Seita** [† 1]

## Abstract

AI assistants in human–AI collaboration often correct suboptimal human actions through behavioral feedback (e.g., alerts or steering-wheel nudges in assistive driving). Such interventions can mitigate immediate errors, but long-term improvement requires addressing the underlying misconceptions that cause repeated mistakes. We introduce SENSEI, a framework that infers user misconceptions from interaction behavior and provides targeted, minimal yet sufficient suggestions to correct them. Our approach departs from action- or trajectory-level interventions by operating over a structured knowledge representation to localize and correct the sources of erroneous behavior. Across three long-horizon tasks with diverse misconceptions and corresponding behaviors, SENSEI demonstrates zero-shot compositional generalization, disentangling multiple overlapping misconceptions despite training only on single-misconception cases. A user study further shows that our method identifies real human misconceptions and provides effective guidance that improves long-horizon task performance, successfully correcting $90\%$ of student misconceptions. Code and project page are available at https://misoshiruseijin.github.io/SENSEI/.

## 1. Introduction

Research in the Learning Sciences emphasizes that robust mastery requires more than correcting individual mistakes; it requires correcting the underlying misconceptions that generate *systematic* errors (Brown & VanLehn, 1980; Gusukuma et al., 2018a; Kennedy et al., 2020; Elmadani et al., 2012). We adopt the same principle to AI assistance for long-horizon decision-making tasks by updating the human's task knowledge so their future behavior matches an expert's, including on new task instances.

Symbolic and model-based work in human–robot collaboration and explainable planning provides an appealing substrate for such adoption, since task knowledge can be represented explicitly and operationalized into behavior through planning (Chakraborti et al., 2017). However, much of prior literature is not posed as knowledge-aware assistance: it typically assumes a known human model (or a finite set of candidate models) and focuses on coordination or behavior explanation, rather than inferring misconceptions from observed behaviors and proposing minimal knowledge corrections (Carroll et al., 2019; Kedia et al., 2024; Losey & Sadigh, 2019; Liang et al., 2024; Chakraborti et al., 2017).

On the other hand, several existing assistive AI systems observe human behavior traces and provide corrections by steering their actions (e.g., alerts, overrides, or guidance) without identifying the misconception underlying the failure (Reddy et al., 2018; Bragg & Brunskill, 2019; Srivastava et al., 2022; Hong et al., 2023). While effective for immediate correction, steering does not directly produce reusable diagnoses of what the human misunderstood, nor does it target interventions that minimize post-intervention errors on future actions in long-horizon settings. In short, prior work often fixes the *move* (via shared autonomy) or fixes the human's understanding of the robot (via explanation), but does not fix the human's *mind* regarding the task.

In this paper, we frame AI-assistance for long-horizon decision-making as *knowledge correction*: aiming to produce targeted guidance that updates a human's internal task knowledge so their post-guidance behavior aligns with an expert in future tasks in a long-horizon setting.

Consider a novice cook who attempts to make a boiled egg by placing a raw egg directly inside a microwave. A behavior-steering assistant might say: "Stop. Use a pot to boil the egg instead." This avoids the immediate disaster, but the user may still retain the same flawed belief about what is safe to microwave. In contrast, knowledge correction

---

*Equal contribution †Equal advising [1]Thomas Lord Department of Computer Science, University of Southern California, Los Angeles, USA. Correspondence to: Ayano Hiranaka <ahiranak@usc.edu>, Ya-Chuan Hsu <yachuanh@usc.edu>.

*Proceedings of the 43$^{rd}$ International Conference on Machine Learning*, Seoul, South Korea. PMLR 306, 2026. Copyright 2026 by the author(s).

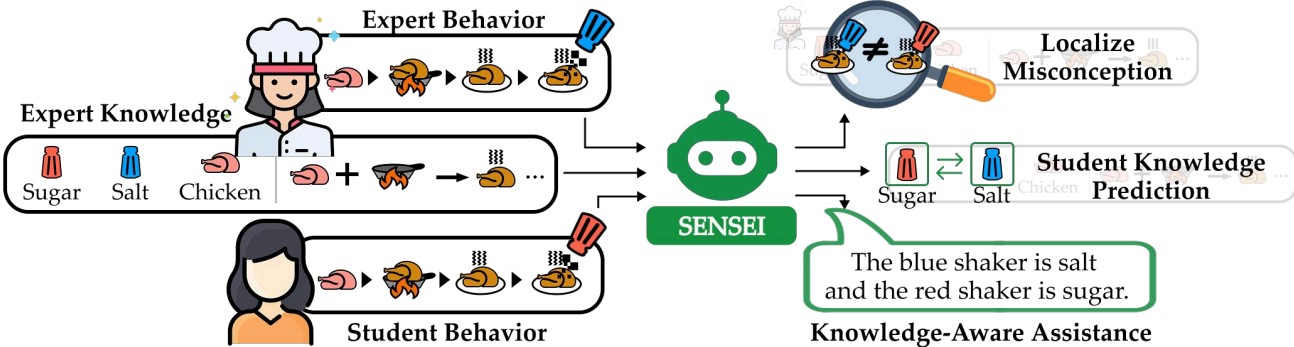

Figure 1. Conceptual overview. Given expert task knowledge, an expert trajectory, and a student trajectory, SENSEI infers the student's underlying knowledge by (i) localizing which symbolic knowledge components are misaligned with the expert (e.g., misconception in item types) and (ii) predicting the student's corresponding variant (e.g., swapping sugar and salt). It then produces a minimal, knowledge-aware assistance (shown in natural language) to correct the misconception and improve task behavior.

targets the misconception: "Microwaving a sealed, liquid-filled shell can cause pressure buildup and explosion." This updated knowledge transfers: the novice cook now knows to vent sealed containers, pierce whole potatoes, and avoid similar hazards in new situations.

We make the following contributions toward knowledge-aware assistance in long-horizon decision-making tasks:

- **Formulation of Knowledge-Aware Assistance.** We formalize assistance as improving post-guidance performance on future tasks by aligning student knowledge to expert knowledge. By targeting understanding of state-independent knowledge rather than context-dependent actions, the formulation supports guidance that generalizes beyond the immediate episode.

- **Framework for Misconception Diagnosis via Structured Corrections.** We propose SENSEI, which identifies task knowledge misconceptions from behavior traces and produces source-identified, interpretable corrections to the human's symbolic knowledge model. This structured diagnose-then-edit design constrains the editor to minimal changes, reducing spurious updates and keeping corrections interpretable.

- **Empirical Validation & Real User Utility.** We evaluate SENSEI on three complex planning domains, demonstrating zero-shot compositional generalization, identifying multiple overlapping misconceptions despite training only on isolated errors. Finally, we conduct a user study with 20 users demonstrating that SENSEI can diagnose and correct real human misconceptions, supporting its practical utility.

## 2. Related Works

Overall, our work lies at the intersection of misconception diagnosis in intelligent tutoring, procedural/physical

assistance, and knowledge-aware human–AI collaboration, combining behavioral evidence with minimal, interpretable task-knowledge updates.

**Intelligent Tutoring Systems (ITS) and Misconception Diagnosis.** Classic ITS track student mastery over predefined skills with methods such as BKT (Corbett & Anderson, 1994), PFA (Pavlik Jr et al., 2009), and their deep learning variants (Piech et al., 2015). While well-suited for predicting future performance, they typically do not recover the specific concept the student is missing or misapplying. A complementary line of work targets *misconceptions*: Repair Theory models errors as systematic application of "buggy" procedures (Brown & VanLehn, 1980), and classroom systems show that correcting misconceptions improves learning (Gusukuma et al., 2018b; Kennedy et al., 2020). However, these approaches often rely on large annotated student datasets (Elmadani et al., 2012) or instructor-authored bug libraries (Gusukuma et al., 2018a).

Recent generative assistants reduce data requirements by proposing fixes to student artifacts (e.g., code/workflows) via learned edit models (Heickal & Lan, 2025) or foundation-model reasoning (Yan et al., 2025; Phung et al., 2026), but the underlying misconception often remains implicit, limiting transfer beyond the immediate artifact. In contrast, SENSEI targets long-horizon decision-making tasks and explicitly infers a structured, reusable misconception correction.

**Assistance in Procedural and Physical Domains.** AI assistance in task-oriented domains often prioritizes immediate performance through action- or trajectory-level interventions. In continuous control (e.g., teleoperation or driving), this typically takes the form of shared autonomy or real-time corrective feedback that steers user actions toward an expert policy (Reddy et al., 2018; Bragg & Brunskill, 2019; Gopinath et al., 2025; Hsu et al., 2026). Even when framed as teaching (e.g., motor skill instruction), these methods

commonly intervene at the level of execution and are evaluated by improved control performance (Srivastava et al., 2022), rather than producing an explicit, reusable diagnosis of the student's underlying misconception.

In higher-level sequential decision-making, related work studies influence and coordination by nudging human partners to improve joint outcomes (Hong et al., 2023). Such systems can substantially improve task efficiency online, but their interventions are typically not designed to yield a structured diagnosis of which aspects of the task the human misunderstood, limiting transfer beyond the current episode or context. In contrast, SENSEI targets knowledge alignment in procedural planning: it infers and corrects the faulty concepts underlying behavioral errors, producing interpretable knowledge-level edits that complement action-level assistance in long-horizon tasks.

**Human Mental Models for AI Assistance.** Knowledge-aware AI assistive frameworks explicitly model what a human knows or believes during joint activity, enabling task and communication actions that maintain common ground (Alami et al., 2006). Epistemic task planning and theory-of-mind approaches track per-agent belief states under partial observability and perspective-taking, and act to prevent or repair belief divergence (Favier et al., 2023; Shekhar et al., 2024). Related work also models perceptual constraints and plans information-revealing actions online by trading off task progress with reducing the human's knowledge gaps (Hsu et al., 2025). While effective for coordination, these methods primarily treat mismatch as state uncertainty and typically rely on a hand-specified human update model, rather than diagnosing which task concepts the human misunderstands.

A complementary symbolic lens comes from Explainable AI Planning, which represents mental-model mismatch as differences between planning models. However, these frameworks (Chakraborti et al., 2017; Sreedharan et al., 2018; Chakraborti et al., 2019) are typically used to explain or optimize robot behavior given assumed model differences, and do not solve the inverse diagnostic problem of inferring a specific, unknown knowledge model from behavioral traces. In contrast, SENSEI targets the inverse setting by localizing faulty symbolic knowledge components and synthesizing minimal corrections that account for the observed deviations.

## 3. Problem Setting: Behavior Alignment via Knowledge Alignment

We study long-horizon decision-making tasks where an expert reliably succeeds, but a human fails *systematically* due to missing or incorrect task knowledge. Our goal is behavior alignment: improving the human's behavior on *future task*

*instances* toward the expert's, while producing interpretable diagnoses of what the human misunderstands (see Fig. 1 for the problem concept).

**Behavior Alignment.** Let $\mathcal{T}_j \sim \mathcal{T}$ be a task instance. The expert has a knowledge representation $K^E$ inducing behavior $\tau^E \sim \pi(\cdot \mid \mathcal{T}_j, K^E)$. The human acts under an unknown internal knowledge $K^S$, yielding $\tau^S \sim \pi(\cdot \mid \mathcal{T}_j, K^S)$. We adopt the standard assumption that an agent acts rationally with respect to their internal model $K$ to complete the task. The assistive AI observes an interaction history $h$ (i.e., $\mathcal{T}_j$ and $\tau^S$; optionally $\tau^E$) and outputs a guidance $u \in \mathcal{U}$ (feedback, explanation, patch, demonstration). The human updates via an unknown mechanism $K^{S+} = \text{Update}(K^S, u)$, then acts again. The assistive objective is to choose a policy $\mu : h \times K^E \mapsto u$ minimizing expected post-guidance regret over future tasks $\mathcal{T}_j$:

$$\min_{\mu} \ \mathbb{E}_{\mathcal{T}'_j \sim \mathcal{T}} \left[ \text{Regret}\big(\pi(\cdot \mid \mathcal{T}'_j, K^{S+}), \ \pi(\cdot \mid \mathcal{T}'_j, K^E)\big) \right],$$

$$(1)$$

where Regret can be instantiated as success gap, cost gap, or trajectory divergence.

Directly optimizing this objective typically requires repeated interventions and long-term outcomes. We therefore approximate the *behavior alignment* objective with a more structured *knowledge alignment* objective.

**Knowledge Components.** To formalize knowledge alignment, assume that knowledge factorizes into interpretable components $\mathcal{K} = \{\phi_0, \dots, \phi_N\}$, where each component $\phi_i$ captures an aspect of world dynamics that is relevant to the task $\mathcal{T}_j$ (e.g., available objects or the feasibility and potential outcomes of an action). For each component index $i$, an agent instantiates a *variant* $v_i \in \mathcal{V}_i$, representing one possible understanding of $\phi_i$. For example, if $\phi_i$ encodes knowledge about washing dishes, one variant may be "I can clean the pan without damaging it," while another variant may be "I know how to clean the pan, but don't know some cleaners may cause damage." A student knowledge model is the resulting configuration $K^S = \{v_i^S\}_{i=0:N}$, while the expert holds $K^E = \{v_i^E\}_{i=0:N}$. We define the *Knowledge Gaps* as the set of component–variant pairs where the student's variant differs from the expert's: $\mathcal{G}(K^E, K^S) = \{(\phi_i, v_i^S) \mid v_i^S \neq v_i^E\}$. A student is fully masterful if $\mathcal{G}(K^E, K^S) = \emptyset$.

**Problem Objectives.** The main objective of the problem is to align the human's behavior in *future task instances* to the expert's behavior. First, given observations $\mathcal{O} = (\mathcal{T}_j, K^E, \tau^S)$ (and optionally $\tau^E$), we first predict the student knowledge $\hat{K}^S = f(\mathcal{O})$ that maximizes the likelihood of the observed student behavior:

$$\max_{\hat{K}^S} P(\tau^S \mid \hat{K}^S) \tag{2}$$

① Knowledge Gap Localization          ② Latent Knowledge Editing and Decoding

*Figure 2.* SENSEI architecture. ① *Knowledge Gap Localization Module*: inputs are embedded by a frozen CodeT5+ encoder $\epsilon$ and the *Localization Network* $f_{\mathrm{loc}}$ predicts which knowledge components a student has gaps in. ② *Knowledge Edit Module*: the *Latent Knowledge Editor Network* $f_{\mathrm{edit}}$ applies latent edits to expert knowledge components to generate a student knowledge component prediction, which is then translated to raw text describing the student knowledge by the *Student Knowledge Decoder*.

Equivalently, this aims to identify the correct set of knowledge gaps $\mathcal{G}(K^E, \hat{K}^S)$.

Conditioned on the predicted knowledge gaps $\mathcal{G}(K^E, \hat{K}^S)$, the final goal is to generate a guidance signal $u$ (e.g., an explanation or instruction) to update the student's knowledge. The objective is to select $u$ such that the student's knowledge gaps are eliminated in the subsequent task instance:

$$\min_{u} |\mathcal{G}(K^E, \mathrm{Update}(\hat{K}^S, u))| \quad (3)$$

where $\mathrm{Update}(\hat{K}^S, u)$ indicates the student knowledge after receiving guidance $u$. Optimizing Eq. (3) is not equivalent to optimizing Eq. (1) as knowledge components have non-uniform behavioral impact. However, if $\mathcal{G}(K^E, K^S) = \emptyset$, then $\mathrm{Regret} = 0$. Thus, knowledge alignment provides a structured proxy objective for behavior alignment.

## 4. Method

Our method, **S**tructured **E**xtraction and **N**eural **S**ynthesis of **E**rrors for **I**ntervention (**SENSEI**), follows a two-stage inference pipeline (Fig. 2). Given $(K^E, \tau^E, \tau^S)$, SENSEI outputs (i) a predicted gap index set $\hat{\mathcal{I}}^S$ from the *Knowledge Gap Localization Module*, where the true gaps are $\mathcal{I}^S = \{i \in \{0, \ldots, N\} \mid v_i^S \neq v_i^E\}$, and (ii) for each $i \in \hat{\mathcal{I}}^S$, an interpretable knowledge block $\hat{v}_i^S$ from the *Knowledge Edit Module*, describing the student's variant for that component.

**Knowledge Representation.** While our problem formulation allows flexibility in knowledge representation, we represent expert and student task knowledge (i.e., $K^E$ and $K^S$) using Planning Domain Definition Language (PDDL), a standard human-readable language for defining planning problems (McDermott et al., 1998). Each PDDL comprises code blocks, such as `action` blocks that define the preconditions and effects of an agent's interaction with the environment, and `objects` blocks that define the types of objects in the environment. We treat each PDDL code block as a variant $v_i$ of the knowledge component $\phi_i$, capturing semantically meaningful information relevant to the task, enabling component-level localization and correction. To

represent agent behaviors ($\tau$), we utilize the symbolic plan solutions to the PDDL problem.

### 4.1. Two-Stage Inference

We encode expert knowledge components and behaviors with a pretrained, frozen CodeT5+ encoder $f_{\mathrm{enc}}$ (Wang et al., 2023). For each expert component, we obtain a component embedding $e_i^E = f_{\mathrm{enc}}(v_i^E)$, where $v_i^E \in K^E$. We also encode expert and student behaviors as $z^E = f_{\mathrm{enc}}(\tau^E)$ and $z^S = f_{\mathrm{enc}}(\tau^S)$, and compute their difference $\Delta z = z^E - z^S$. Additionally, we encode the full expert knowledge $K^E$ with $f_{\mathrm{enc}}$ (via concatenating its components), to obtain a global knowledge embedding $e_G$.

**Stage 1: Knowledge Gap Localization.** The localization network $f_{\mathrm{loc}}$ predicts which components are inconsistent with the student's behavior. Concretely, for each component $i \in \{0, \ldots, N\}$, $f_{\mathrm{loc}}$ produces a logit $\ell_i$ (and binary label $\hat{g}_i \in \{0, 1\}$) predicting whether $i \in \mathcal{I}^S$:

$$\ell_i = f_{\mathrm{loc}}(z^E, z^S, \Delta z, e_i^E, e_G), \quad \hat{g}_i = \mathbb{1}[\sigma(\ell_i) > 0.5], \quad (4)$$

where $\sigma(\cdot)$ is the sigmoid function.

Localizing at the component level keeps computation per component fixed. Regardless of the total number of knowledge components or domain size, each prediction only processes a single component embedding $e_i^E$ together with trajectory summaries. Such a design decouples model complexity from knowledge complexity and scales to large domains without requiring larger networks.

**Stage 2: Latent Knowledge Editing and Decoding.** Given the localization outputs, the latent editor $f_{\mathrm{edit}}$ predicts an edit vector $\Delta e_i$ for each component and forms a predicted student embedding by editing the corresponding expert embedding:

$$\hat{e}_i^S = e_i^E + \hat{g}_i \cdot \Delta e_i, \quad \Delta e_i = f_{\mathrm{edit}}(z^E, z^S, \Delta z, e_i^E, e_G). \quad (5)$$

Intuitively, $\hat{e}_i^S$ represents the student's variant $v_i^S$ in latent space for that component. During training, we use a *soft gate* for smoother gradient flow by replacing the hard label

with $\sigma(\ell_i)$:

$$\hat{e}_i^S = e_i^E + \sigma(\ell_i) \cdot \Delta e_i. \qquad (6)$$

To produce an interpretable correction, we decode the edited embedding back into a raw knowledge text block $\hat{v}_i^S$ using a CodeT5+ decoder $f_{\text{dec}}$ finetuned to reconstruct text from knowledge embeddings. This design structurally separates localization from correction, constraining the editor to act only on predicted gaps. As a result, it reduces hallucinated edits to correct components of the student knowledge, keeping the correction minimal and interpretable.

### 4.2. Training Objectives

Training targets three outcomes aligned with our evaluation metrics: (i) accurate gap localization over components, (ii) accurate gap correction on true-gap components, and (iii) minimal/no edits on non-gap components. We train all trainable modules jointly, while ensuring each loss updates only its intended sub-network as described below.

**Localization Network ($f_{\text{loc}}$).** We train $f_{\text{loc}}$ to predict the gap indicator $\hat{g}_i$ with a primary Single-Misconception Localization Loss $\mathcal{L}_{\text{single}}$ and an auxiliary Mixup Generalization Loss $\mathcal{L}_{\text{mix}}$:

$$\mathcal{L}_{\text{loc}} = \mathcal{L}_{\text{single}} + \lambda_{\text{mix}}\mathcal{L}_{\text{mix}}. \qquad (7)$$

Both terms use binary cross-entropy (BCE); $\mathcal{L}_{\text{single}}$ is computed from embeddings of single-misconception behaviors, while $\mathcal{L}_{\text{mix}}$ regularizes $f_{\text{loc}}$ to generalize to multi-misconception inputs, whose behavior embeddings exhibit a distribution shift due to interacting gap signals.

To construct pseudo multi-misconception training examples, we synthesize behavior embeddings from two students $A$ and $B$ with gap index sets $\mathcal{I}^A$ and $\mathcal{I}^B$:

$$z^{\text{mix}} = \alpha z^A + (1-\alpha)z^B, \qquad \alpha \sim U[0,1]. \qquad (8)$$

We supervise predictions from $z^{\text{mix}}$ with the *union* label $\mathcal{I}^{\text{mix}} = \mathcal{I}^A \cup \mathcal{I}^B$, i.e., $\mathcal{L}_{\text{mix}}$ is a BCE loss using $\mathcal{I}^{\text{mix}}$ as ground truth. Intuitively, this encourages $f_{\text{loc}}$ to recover each component-specific gap signal even when the embedding is perturbed by other misconceptions, thereby simulating interference patterns in real multi-gap trajectories.

**Latent Knowledge Editor Network ($f_{\text{edit}}$).** To ensure $f_{\text{edit}}$ outputs meaningful representations, we train the module using an Alignment Loss ($\mathcal{L}_{\text{align}}$) and a No-Edit Loss ($\mathcal{L}_{\text{no-op}}$):

$$\mathcal{L}_{\text{edit}} = \mathcal{L}_{\text{align}} + \lambda_{\text{no-op}}\mathcal{L}_{\text{no-op}} \qquad (9)$$

where $\mathcal{L}_{\text{align}}$ and $\mathcal{L}_{\text{no-op}}$ are defined as:

$$\mathcal{L}_{\text{align}} = \sum_{i \in \mathcal{I}^S} \text{MSE}(\hat{e}_i^S, e_i^S), \; \mathcal{L}_{\text{no-op}} = \sum_{i \notin \mathcal{I}^S} L_2(\Delta e_i) \quad (10)$$

The mean-squared error (MSE) loss $\mathcal{L}_{\text{align}}$ enforces the predicted student knowledge variant embedding $\hat{e}_i^S$ to align with the ground truth student variant embedding $e_i^S = f_{\text{enc}}(v_i^S)$. The $L_2$ loss $\mathcal{L}_{\text{no-op}}$ discourages the editor to modify components that are not gaps ($g_i = 0$).

**Student Knowledge Decoder Model ($f_{\text{dec}}$).** We first project $\hat{e}_i^S$ into a fixed set of memory tokens $m(\hat{e}_i^S)$, which the decoder attends to when generating text. Let $(w_{i,1}^S, \ldots, w_{i,T_i}^S)$ denote the tokenization of the ground-truth student knowledge. We then train the decoder with teacher forcing using the standard auto-regressive cross-entropy loss on the student's ground-truth gap tokens:

$$\mathcal{L}_{\text{dec}} = -\sum_t \log p(w_{i,t} \mid w_{i,<t}, m(\hat{e}_i)). \qquad (11)$$

See Appendix B for model architecture and training details.

## 5. Experiments

### 5.1. Dataset

Our dataset contains tuples of $(K^E, K^S, \tau^E, \tau^S)$ and ground-truth faulty component labels $\mathcal{I}^S$. We sample student models with 1 to 3 misconceptions, generate plans with OPTIC (Benton et al., 2012), and augment the data with additional valid plan variants by reordering actions while preserving plan validity. Note that we train only on 1 misconception data, reserving 2- and 3- misconception data solely for evaluation. See Appendix A for details.

Specifically, we simulate student models by perturbing the expert PDDL in three ways, chosen to induce qualitatively distinct plan signatures:

- *Type* (item misuse): the agent miscategorizes objects and treats an object as the wrong class. Simulated by swapping type assignments in the `objects` block.

- *Hazard* (risky behavior): the agent ignores negative side-effects, producing plans that look efficient but violate safety/capacity constraints. Simulated by removing negative predicates from action `effects`.

- *Skill* (workarounds): the agent cannot execute a key capability, leading to detours that bypass the missing skill. Simulated by assigning a prohibitive cost to the corresponding action.

The above categories are a starting point, rather than an exhaustive list, to model a full range of real-world human errors. However, SENSEI's architecture is not bound by these categories. Since SENSEI decouples knowledge complexity from model complexity, it scales to complex real-world misconceptions by expanding the PDDL representations and collecting additional human data.

## 5.2. Domains

We evaluate on three planning domains that admit diverse, solvable knowledge perturbations (details in Appendix A.1):

- **Breakfast**: a household cooking domain (ingredients, tools, appliances) where plans involve multi-step preparation and cleanup.

- **Overcooked**: a single-agent variant of Overcooked-AI (Carroll et al., 2019) with navigation, ingredient processing, and dish assembly in a kitchen layout that constrains paths and reachability (e.g., locked doors force longer routes to stations).

- **Rover**: a modified International Planning Competition's Rovers task (Long & Fox, 2003) where a rover acquires soil/rock samples and images and transmits results. We add a capacity/communication structure to produce diverse behaviors under misconceptions.

## 5.3. Baselines

To the best of our knowledge, SENSEI is the first framework to address the problem of interpretable knowledge diagnosis through explicit correction of structured symbolic knowledge representations. As no off-the-shelf methods exist for this framework, we construct baselines representing different plausible approaches to our problem:

**Monolithic End-to-End Model.** We train a single conditional decoder that generates the corrected PDDL block from embeddings of the expert knowledge components and behaviors, together with the student's behavior. A frozen CodeT5+ encoder produces $(z^E, z^S, \Delta z, e_i^E)$, which are projected into a fixed set of memory tokens. A CodeT5+ decoder then cross-attends to decode $\hat{v}_i^S$ with teacher forcing. This baseline has no explicit gap localizer or editor; both whether and how to edit must be learned implicitly. For fairness, we initialize from the same pretrained decoder and projection checkpoints as our main method and fine-tune on component-level corrections.

**Off-the-Shelf LLMs.** To investigate if specialized training is needed, we evaluate a general-purpose reasoning baseline using GPT-4o and GPT-5.2 (Achiam et al., 2023), and Llama-3.3-70B (Grattafiori et al., 2024). We prompt each model with expert knowledge and behavior, student behavior, and natural language instructions to identify gap components and propose corrections that reflect the student's misconceptions. Conditions $\hat{e}_i^S = e_i^S$ and $\hat{e}_i^S = e_i^E$ were judged by prompting GPT-4o to evaluate the semantic equivalence of the student PDDL blocks generated by the LLM and true target blocks. Additional details are in Appendix E.

**Additional Localization Bounds.** In addition to the above baselines, we evaluate localization bounds that isolate sources of misconceptions: (i) *Random-Loc*, which selects components uniformly at random at each trajectory step; (ii) *Random-H-Loc*, which samples uniformly from components that appeared as gaps in the training set; and (iii) *Oracle-Loc*, which returns the ground-truth gap indices.

## 5.4. Evaluation Metrics

We evaluate SENSEI at both the *component* level (gap localization vs. edit module) and the *system* level (full framework). All metrics are computed over PDDL blocks. Let $g_i \in \{0, 1\}$ indicate whether component $i$ contains a ground-truth gap, and let $\hat{g}_i \in \{0, 1\}$ be the predicted gap indicator to trigger a correction. If $\hat{g}_i=1$, the edit module outputs an edited student component $\hat{v}_i^S$. We compare $\hat{v}_i^S$ to the true student component $v_i^S$ if the localization is correct ($g_i = 1$) or the expert component $v_i^E$ if the localization is erroneous ($g_i = 0$). We annotate each metric with $\uparrow$ (higher is better) or $\downarrow$ (lower is better).

**Localization (where to correct).** *Localization Recall* (Loc_rec) measures how many true gaps are localized, while *Localization Precision* (Loc_prec) measures how often a localized gap corresponds to a true gap:

$$\text{Loc}_{\text{rec}} (\uparrow) = \frac{\sum_i \mathbb{1}[g_i=1 \land \hat{g}_i=1]}{\sum_i \mathbb{1}[g_i=1]},$$

$$\text{Loc}_{\text{prec}} (\uparrow) = \frac{\sum_i \mathbb{1}[g_i=1 \land \hat{g}_i=1]}{\sum_i \mathbb{1}[\hat{g}_i=1]}.$$

**Edit (correction quality given a trigger).** *Edit Accuracy on Correct Localizations* (Edit_acc) measures how often we accurately correct a localized true gap, and *No-Op Rate on Erroneous Localizations* (Edit_no-op) measures how often we avoid damaging a correct component under an incorrect localization:

$$\text{Edit}_{\text{acc}} (\uparrow) = \frac{\sum_i \mathbb{1}[g_i=1 \land \hat{g}_i=1 \land \hat{e}_i^S=e_i^S]}{\sum_i \mathbb{1}[g_i=1 \land \hat{g}_i=1]},$$

$$\text{Edit}_{\text{no-op}} (\uparrow) = \frac{\sum_i \mathbb{1}[g_i=0 \land \hat{g}_i=1 \land \hat{e}_i^S=e_i^E]}{\sum_i \mathbb{1}[g_i=0 \land \hat{g}_i=1]}.$$

**System (overall utility).** *System Recall* (Sys_rec) measures what fraction of all true gaps are successfully corrected by the full pipeline, *System Precision* (Sys_prec) measures how often a correction is both necessary and successful, and *System False Correction Rate* (Sys_false) measures how often the system incorrectly localizes and modifies a no-gap

component:

$$\text{Sys}_{\text{rec}} (\uparrow) = \frac{\sum_i \mathbb{1}[g_i=1 \wedge \hat{g}_i=1 \wedge \hat{e}_i^S=e_i^S]}{\sum_i \mathbb{1}[g_i=1]},$$

$$\text{Sys}_{\text{prec}} (\uparrow) = \frac{\sum_i \mathbb{1}[g_i=1 \wedge \hat{g}_i=1 \wedge \hat{e}_i^S=e_i^S]}{\sum_i \mathbb{1}[\hat{g}_i=1]},$$

$$\text{Sys}_{\text{false}} (\downarrow) = \frac{\sum_i \mathbb{1}[g_i=0 \wedge \hat{g}_i=1 \wedge \hat{e}_i^S \neq e_i^E]}{\sum_i \mathbb{1}[g_i=0]}.$$

## 5.5. Experiment Results

To evaluate performance, we report the F1 scores for localization and end-to-end correction ($\text{Loc}_{\text{F1}}$ and $\text{Sys}_{\text{F1}}$, highlighted in blue in Table 1) as summary metrics capturing the precision–recall trade-off inherent to diagnosis. We also measure correction accuracy on localized true-gap components ($\text{Edit}_{\text{acc}}$). For interpretability, we report $\text{Loc}_{\text{rec}}$, $\text{Loc}_{\text{prec}}$ and $\text{Sys}_{\text{rec}}$, $\text{Sys}_{\text{prec}}$, along with $\text{Sys}_{\text{false}}$ to capture erroneous corrections to components with no true gaps. To isolate correction-synthesis behavior, we report the no-op rate on erroneously localized no-gap components ($\text{Edit}_{\text{no-op}}$).

**Overall Performance.** In all domains, SENSEI achieves the best balance between correcting true gaps and avoiding erroneous corrections: Breakfast ($\text{Sys}_{\text{F1}} = 0.709$), Overcooked ($0.726$), and Rover ($0.561$). In contrast, most learned baselines either miss the majority of gaps (low $\text{Sys}_{\text{rec}}$) or produce overly broad diagnoses (low $\text{Sys}_{\text{prec}}$), resulting in low $\text{Sys}_{\text{F1}}$.

**Localization is Non-Trivial.** Random localization baselines achieve high system recall by identifying many components as gaps, but their low precision leads to poor $\text{Sys}_{\text{F1}}$ (e.g., Random-Loc: $0.069$ in Overcooked) and high erroneous edit rates ($\text{Sys}_{\text{false}} = 0.957–1.0$). SENSEI substantially improves localization quality, achieving consistently higher $\text{Loc}_{\text{F1}}$ (Breakfast/Overcooked/Rover: $0.731/0.726/0.561$) than Random-H-Loc ($0.369/0.500/0.299$), indicating it selects a smaller, more relevant set of components for correction without collapsing recall.

**End-to-End vs. Modular Correction.** The End-to-End baseline significantly underperforms SENSEI in localizing the gap components (e.g., Overcooked $\text{Loc}_{\text{rec}} = 0.042$; Breakfast $\text{Loc}_{\text{rec}} = 0.291$). The correction synthesis is also unreliable as the editor tends to edit knowledge components regardless of necessity across all three domains, $\text{Edit}_{\text{no-op}} = 0.0$. This highlights the benefits of decoupling *where to edit* from *how to edit*, as SENSEI maintains strong localization capabilities while producing highly accurate corrections ($\text{Edit}_{\text{acc}} = 0.97–1.0$), yielding consistently higher $\text{Sys}_{\text{F1}}$.

**LLM Baselines.** Direct LLM-based correction struggles with producing consistent PDDL corrections, reflected by high abstention/no-op rates (e.g., $\text{Edit}_{\text{no-op}} = 0.539–0.760$) and low localization recall, yielding low end-to-end perfor-

mance (especially in Rover where $\text{Sys}_{\text{F1}} = 0$). This gap motivates our structured editor, which achieves near-perfect correction accuracy once the appropriate component is localized (e.g., $\text{Edit}_{\text{acc}} \geq 0.97$ for SENSEI).

## 5.6. Ablation Studies

To validate our architectural design, we analyze the contribution of the mixup generalization loss and the expert reference. Results are presented in Table 2.

**Effect of Latent Mixup in Generalization.** We train a variant No-Mixup without the auxiliary mixup generalization loss ($\lambda_{\text{mix}} = 0$). This variant suffers a catastrophic drop in system recall, particularly in the Breakfast domain (drop from $0.762$ to $0.221$). Without the synthetic mixed behaviors, the localization network becomes extremely conservative, validating $\mathcal{L}_{\text{mix}}$ as a significant contributor to SENSEI's zero-shot compositional generalization capabilities.

**The Role of Expert Reference.** Although No-Expert achieves higher $\text{Sys}_{\text{F1}}$ than SENSEI in Breakfast and Rover, the underlying metrics indicate a different precision-recall trade-off rather than a failure of expert anchoring. As shown in Table 2, No-Expert yields lower recall and higher precision than SENSEI. Replacing the expert anchor with a randomly sampled student during training exposes the model to behavioral differences between two suboptimal, noisy agents. We hypothesize that this encourage a more conservative decision boundary: the model requires stronger behavioral evidence before declaring knowledge gaps. As a result, No-Expert produces fewer false positives and achieves higher precision, but misses more subtle gaps, reducing recall. Conversely, SENSEI (expert-anchored during training and inference) is highly sensitive to deviations from expert-aligned knowledge and yields higher recall across all three domains. This distinction also helps explain the threshold-tuning results in Appendix C.1. When each method is optimized for $\text{Sys}_{\text{F1}}$, SENSEI achieves the highest average $\text{Sys}_{\text{F1}}$ across the three tasks. Ultimately, No-Expert's competitive performance is encouraging, demonstrating our framework's potential for general agent-agent alignment where a definitive expert is unavailable.

## 5.7. User Study

**Experimental Setup.** We conducted a user study with 20 users in our modified Overcooked task (Fig. 3). Users were assigned to one of four groups. All groups shared the same goal, but we induced two misconceptions per group by withholding key instructions (e.g., how to open a locked door). Users first attempted the task with incomplete understanding, then reattempted after receiving SENSEI's targeted natural-language guidance. Additional information is provided in Appendix D. The study was approved by an Institutional Review Board.

*Table 1.* Simulation Experiment Results. Bold indicates the best values; blue highlights indicate the main metrics we use for evaluation; the "–" indicates that the metric is not applicable, since no erroneous localization on true gaps occurred. See Sec. 5.5 for more details.

| | | $\text{Loc}_{rec}(\uparrow)$ | $\text{Loc}_{prec}(\uparrow)$ | $\text{Loc}_{F1}(\uparrow)$ | $\text{Edit}_{acc}(\uparrow)$ | $\text{Edit}_{no\text{-}op}(\uparrow)$ | $\text{Sys}_{rec}(\uparrow)$ | $\text{Sys}_{prec}(\uparrow)$ | $\text{Sys}_{F1}(\uparrow)$ | $\text{Sys}_{false}(\downarrow)$ |
|---|---|---|---|---|---|---|---|---|---|---|
| | Random-Loc | **0.997** | 0.104 | 0.188 | 0.875 | 0.0 | **0.875** | 0.092 | 0.166 | 0.999 |
| | Random-H-Loc | **0.998** | 0.226 | 0.369 | 0.876 | 0.0 | **0.876** | 0.199 | 0.324 | 0.400 |
| | End-to-End | 0.424 | 0.514 | 0.465 | 0.947 | 0.0 | 0.401 | 0.487 | 0.440 | 0.043 |
| Breakfast | GPT-4o | 0.147 | 0.339 | 0.205 | 0.524 | 0.720 | 0.077 | 0.177 | 0.107 | 0.007 |
| | GPT-5.2 | 0.157 | 0.257 | 0.195 | 0.644 | 0.392 | 0.101 | 0.166 | 0.126 | 0.024 |
| | Llama-3.3-70B | 0.118 | 0.203 | 0.149 | 0.059 | 0.040 | 0.007 | 0.013 | 0.009 | 0.037 |
| | **SENSEI (ours)** | 0.786 | 0.683 | **0.731** | **0.97** | 0.0 | 0.762 | **0.663** | **0.709** | 0.043 |
| | Oracle-Loc (Upper) | 1.0 | 1.0 | 1.0 | 0.867 | 0.0 | 0.867 | 0.867 | 0.867 | 0.0 |
| | Random-Loc | **1.0** | 0.040 | 0.077 | 0.925 | 0.0 | **0.925** | 0.036 | 0.069 | 1.0 |
| | Random-H-Loc | **1.0** | 0.333 | 0.500 | 0.925 | 0.0 | **0.925** | 0.308 | 0.462 | 0.082 |
| | End-to-End | 0.042 | 0.027 | 0.033 | 0.853 | 0.0 | 0.036 | 0.023 | 0.028 | 0.061 |
| Overcooked | GPT-4o | 0.185 | 0.196 | 0.190 | 0.027 | 0.539 | 0.005 | 0.005 | 0.005 | 0.012 |
| | GPT-5.2 | 0.275 | 0.224 | 0.247 | 0.309 | 0.220 | 0.085 | 0.069 | 0.076 | 0.026 |
| | Llama-3.3-70B | 0.335 | 0.179 | 0.233 | 0.104 | 0.300 | 0.035 | 0.019 | 0.025 | 0.038 |
| | **SENSEI (ours)** | 0.828 | 0.646 | **0.726** | **1.0** | 0.0 | 0.828 | **0.646** | **0.726** | 0.019 |
| | Oracle-Loc (Upper) | 1.0 | 1.0 | 1.0 | 0.924 | 0.0 | 0.924 | 0.924 | 0.924 | 0.0 |
| | Random-Loc | **1.0** | 0.088 | 0.162 | 0.940 | 0.040 | 0.939 | 0.083 | 0.153 | 0.957 |
| | Random-H-Loc | **1.0** | 0.176 | 0.299 | 0.940 | 0.090 | **0.942** | 0.166 | 0.282 | 0.410 |
| | End-to-End | 0.291 | 0.424 | 0.176 | 1.0 | 0.0 | 0.291 | 0.424 | 0.176 | 0.038 |
| Rover | GPT-4o | 0.008 | 0.016 | 0.011 | 0.0 | 0.760 | 0.0 | 0.017 | 0.0 | 0.0 |
| | GPT-5.2 | 0.008 | 0.016 | 0.011 | 0.0 | 0.184 | 0.0 | 0.056 | 0.0 | 0.0 |
| | Llama-3.3-70B | 0.012 | 0.021 | 0.015 | 0.0 | 0.072 | 0.0 | 0.071 | 0.0 | 0.0 |
| | **SENSEI (ours)** | 0.726 | 0.457 | **0.561** | **1.0** | 0.03 | 0.727 | **0.457** | **0.561** | 0.081 |
| | Oracle-Loc (Upper) | 1.0 | 1.0 | 1.0 | 0.935 | 0.0 | 0.935 | 0.935 | 0.935 | 0.0 |

*Table 2.* Ablation Experiment Results.

| | | $\text{Sys}_{rec}(\uparrow)$ | $\text{Sys}_{prec}(\uparrow)$ | $\text{Sys}_{F1}(\uparrow)$ |
|---|---|---|---|---|
| | No-Mixup | 0.221 | 0.954 | 0.359 |
| Breakfast | No-Expert-Traj | 0.673 | 0.793 | 0.728 |
| | No-Expert | 0.635 | 0.923 | 0.752 |
| | **SENSEI** | **0.762** | 0.663 | 0.709 |
| | No-Mixup | 0.690 | 0.707 | 0.698 |
| Overcooked | No-Expert-Traj | 0.774 | 0.670 | 0.718 |
| | No-Expert | 0.747 | 0.672 | 0.708 |
| | **SENSEI** | **0.828** | 0.646 | **0.726** |
| | No-Mixup | 0.457 | 0.743 | 0.566 |
| Rover | No-Expert-Traj | 0.673 | 0.365 | 0.473 |
| | No-Expert | 0.680 | 0.512 | 0.584 |
| | **SENSEI** | 0.727 | 0.457 | 0.561 |

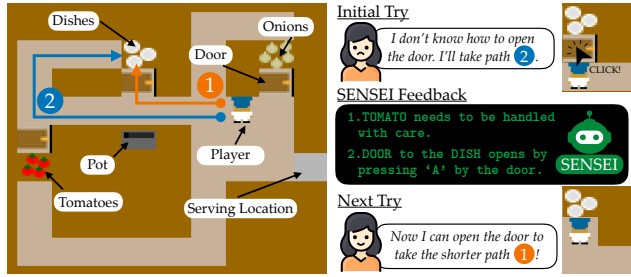

*Figure 3.* User Study Overview. Users completed a soup-cooking task in our modified Overcooked-AI environment by placing onions and tomatoes into the pot, cooking and plating the soup, and delivering it to the serving location. In the initial trial, we withheld key task information (e.g., how to open locked doors). SENSEI then identified the missing knowledge and outputted targeted guidance to a terminal. After reading the guidance, users completed the next trial, improving their play from the initial trial.

**Quantitative Results.** Using the purposely withheld information as the ground truth knowledge gaps, we compute $\text{Sys}_{rec}$ and $\text{Sys}_{prec}$. Additionally, we compute intersection over union (IoU) of plan steps to measure the alignment between the subject and expert behavior before and after guidance. Across 20 subjects, SENSEI corrected knowledge gaps with $\text{Sys}_{rec} = 0.895$ and $\text{Sys}_{prec} = 0.555$, increasing IoU alignment scores from $0.736$ to $0.844$ (10.8% increase).

**Qualitative Results.** Post-study surveys indicate that, per session, users received an average of $1.90$ helpful tips (revealing new information) and $1.64$ unhelpful tips (i.e., SENSEI falsely identified gaps and provided information they already knew). This ratio suggests that while successful diagnoses are highly valued, erroneous corrections remain a key source of friction. Nevertheless, users rated the system's *subjective helpfulness* at $4.27/5$ (Likert), suggesting a positive reception: knowledge-aware guidance is valuable enough that users tolerate frequent unhelpful feedback.

## 6. Discussion and Conclusion

We presented SENSEI, a knowledge-correction framework for AI-assisted long-horizon decision-making. Given an expert reference and student behavior, SENSEI models the student as a planner with a potentially flawed task knowl-

edge, (i) localizes the symbolic knowledge components that explain observed deviations, and (ii) synthesizes minimal, interpretable corrections. Across three domains with diverse misconception types, SENSEI achieves strong end-to-end correction quality and exhibits zero-shot compositional generalization to multi-misconception behaviors despite being trained only on single-misconception trajectories. A user study (20 users) further demonstrates SENSEI identifying human misconception-induced errors and improving post-guidance performance toward expert behavior.

These results also surface important open challenges. Maximizing coverage can increase erroneous corrections, motivating better interaction strategies that reduce unnecessary guidance without sacrificing recall. In addition, since SENSEI is grounded in an expert reference model, it inherits a normative notion of optimality defined by the expert. Applying this approach to open-ended settings would require care to avoid penalizing valid alternative strategies. Future work includes extending beyond PDDL domains and developing value-aware approaches for misconception localization. By weighting knowledge gaps based on their downstream behavioral impact rather than treating them uniformly, we aim to bridge the remaining gap between the knowledge alignment and behavior alignment objectives. Overall, SENSEI supports a shift from trajectory-level steering toward structured knowledge-model correction as a foundation for interpretable AI assistance in long-horizon tasks.

## Impact Statement

This work aims to advance the capabilities of AI assistance systems by enabling interpretable diagnosis of student knowledge misconceptions and providing assistance to resolve the misconceptions. By moving beyond opaque behavior steering toward transparent, logic-based diagnosis, our framework has the potential to democratize access to personalized, high-quality instruction in complex domains.

However, we acknowledge the potential risks associated with automated misconception diagnosis. First, our reliance on expert reference trajectories and pre-defined PDDL domains implies a normative definition of "optimality." In open-ended real-world scenarios, this rigid adherence to a specific expert's logic could inadvertently penalize creative or alternative problem-solving strategies that deviate from the reference but remain valid. Second, as observed in our user study, diagnostic systems are prone to erroneous corrections (hallucinating misconceptions where none exists). In real-world settings, "hallucinating" errors can lead to student frustration, confusion, or decreased self-efficacy. Therefore, we emphasize that such systems should be deployed as supportive tools rather than autonomous evaluators, particularly in high-stakes environments where diagnostic accuracy and precision is critical.

## Acknowledgements

This research was supported in part by research gifts from OpenAI.

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

# A. Domain and Dataset

Evaluating misconception diagnosis on real human behavior is challenging because the student's true internal misconception is often unobserved. We therefore construct controlled planning domains in which we can explicitly generate both (i) different student knowledge models, by perturbing the expert PDDL to introduce known misconceptions, and (ii) different student behaviors, by planning under those perturbed knowledge models and augmenting the resulting plans with diverse valid action permutations. This setup gives us ground-truth knowledge gaps while still exposing SENSEI to behavioral variation across agents and trajectories.

Using established benchmarks (Overcooked, IPC Rover) and a custom Breakfast task, we test compositional generalization of SENSEI across distinct and overlapping misconceptions. Our three misconception categories (*Type*, *Hazard*, *Skill*, as described in Section 5.1) captures common ways symbolic task knowledge can fail: misclassifying object types, ignoring negative effects, or lacking an executable capability. Each misconception was authored and verified by humans to ensure logical consistency.

Each of these misconceptions maps to a unique known set of gap components and induces a distinct family of agent behaviors. To avoid evaluating only a single canonical trajectory per student model, we augment each student's plan with diverse, valid action permutations that preserve task feasibility.

In this section, we provide detailed descriptions of each task and misconceptions, additional dataset details, and an example of PDDL snippets and a plan from a student with two misconceptions.

## A.1. Task Domains

The task domains and misconceptions used to evaluate SENSEI are detailed below:

### A.1.1. BREAKFAST TASK

The agent is tasked to prepare breakfast by cooking an omelet, topping it with tomato sauce, and serving milk in a mug. Objects in the environment include two variants of eggs (raw, boiled), canned tomato sauce, two variants of milk (fresh, spoiled), two variants of bowls (microwave-safe, microwave-unsafe), a plate, a pan, a mug, sugar, salt, and two variants of cleaning supplies (sponge, metal brush). Agents can determine the correct variant of eggs and seasoning (sugar and salt) by taking *verification* actions.

The expert plan is as follows:

```
(verify_is_salt salt1)
(verify_is_raw_egg raw_egg1)
(serve_milk_fresh milk2 cup1)
(crack_egg_raw raw_egg1 bowl2)
(beat_egg raw_egg1 bowl2)
(season_egg_with_salt raw_egg1 salt1 bowl2)
(preheat_pan pan1)
(transfer_ingredient raw_egg1 bowl2 pan1)
(cook_ingredient_in_preheated raw_egg1 pan1)
(plate_egg raw_egg1 pan1 plate1)
(open_can tomatosauce1)
(dump_can_to tomatosauce1 bowl1)
(microwave_ingredient_safe tomatosauce1 bowl1)
(add_topping raw_egg1 tomatosauce1 bowl1)
(clean_container_soft pan1 sponge1)
(clean_container_harsh_damaging bowl2 metal_brush1)
(clean_container_harsh_damaging bowl1 metal_brush1)
(end_task)
```

*Listing 1.* Breakfast expert plan

Below are the descriptions of 11 unique misconceptions simulated for the Breakfast task, categorized as *hazard*, *skill*, or *type* (see Sec. 5.1):

1. Unsafe Microwaving (*Hazard*): Student does not know microwaving a microwave-unsafe bowl causes damage.

2. Harmful Cleaning (*Hazard*): Student does not know that using a metal brush on a pan scratches the pan.

3. Stubborn Sticking (*Hazard*): Student does not know cooking food in a non-preheated pan causes stubborn sticking.

4. No Clean Skill (*Skill*): Student does not know how to clean tools used for cooking.

5. No Crack Egg Skill (*Skill*): Student does not know how to crack eggs.

6. No Open Can Skill (*Skill*): Student does not know how to open a can.

7. No Egg Verify Skill (*Skill*): Student does not know how to check if an egg is raw or boiled without cracking it.

8. No Spice Verify Skill (*Skill*): Student does not know how to check if a seasoning is salt or sugar.

9. Fresh-Spoiled Confusion (*Type*): Student thinks milk1 is fresh and milk2 is spoiled.

10. Salt-Sugar Confusion (*Type*): Student thinks salt1 item is sugar and sugar1 item is salt.

11. Raw-Boiled Confusion (*Type*): Student thinks the raw egg is boiled and the boiled egg is raw.

### A.1.2. OVERCOOKED

The task consists of placing 2 onions and 1 tomato into a pot, cooking the soup, plating the soup, and delivering it to a serving location. Doors in the environment unlock shortcut paths to each item, and the agent must take shortcut paths whenever possible. Additionally, tomatoes and dishes must be handled with care, as they are delicate items.

The expert plan is as follows:

```
(move_to_door_onion p1 start_loc loc7 door1)
(open_door_onion p1 loc7 door1)
(move_to_onion_dispenser_short p1 loc7 loc1 onion_d1 door1)
(pickup_onion_from_dispenser_careless_harmless p1 loc1 onion_d1)
(move_to_pot p1 loc1 loc2 pot1)
(place_onion_in_pot p1 loc2 pot1)
(move_to_onion_dispenser_short p1 loc2 loc1 onion_d1 door1)
(pickup_onion_from_dispenser_careless_harmless p1 loc1 onion_d1)
(move_to_pot p1 loc1 loc2 pot1)
(place_onion_in_pot p1 loc2 pot1)
(move_to_door_tomato p1 loc2 loc8 door2)
(open_door_tomato p1 loc8 door2)
(move_to_tomato_dispenser_short p1 loc8 loc4 tomato_d1 door2)
(pickup_tomato_from_dispenser_careful_needed p1 loc4 tomato_d1)
(move_to_pot p1 loc4 loc2 pot1)
(place_tomato_in_pot p1 loc2 pot1)
(cook_soup p1 loc2 pot1)
(cook_soup p1 loc2 pot1)
(cook_soup p1 loc2 pot1)
(finished_cooking_correct_recipe p1 loc2 pot1)
(move_to_door_dish p1 loc2 loc9 door3)
(open_door_dish p1 loc9 door3)
(move_to_dish_dispenser_short p1 loc9 loc3 dish_d1 door3)
(pickup_dish_from_dispenser_careful_needed p1 loc3 dish_d1)
(move_to_pot p1 loc3 loc2 pot1)
(pickup_correct_soup p1 loc2 pot1)
(move_to_serve_counter p1 loc2 loc5 serve_c1)
(serve_correct_soup p1 loc5 serve_c1)
(end_task)
```

*Listing 2.* Overcooked expert plan

Below are the descriptions of 7 unique misconceptions simulated for the Overcooked task, categorized as *hazard*, *skill*, or *type* (see Sec. 5.1):

1. Careless Dish (*Hazard*): Student does not know that dishes should be picked up carefully.

2. Careless Tomato (*Hazard*): Student does not know that tomatoes should be picked up carefully.

3. No Open Dish Door (*Skill*): Student does not know how to open the door to the dishes.

4. No Open Onion Door (*Skill*): Student does not know how to open the door to the onions.

5. No Open Tomato Door (*Skill*): Student does not know how to open the door to the tomatoes.

6. Onion-Tomato Confusion (*Type*): Student thinks onions are tomatoes and tomatoes are onions.

7. Counter Confusion (*Type*): Student thinks the white counter is the serving counter.

### A.1.3. ROVER

The task is a modified IPC *Rovers* domain where a rover must navigate among waypoints to collect soil and rock samples, take images of scientific targets, and transmit all results back to a lander. Objects include the rover, a set of waypoints, sample targets (soil/rock), image targets, a camera and calibration object, a lab module, and a lander communication endpoint. To induce diverse plan signatures under misconceptions, we add a lightweight capacity/communication structure: the rover has a limited onboard store that must be emptied between samples, and communication requires establishing a link that occupies a shared channel until explicitly released (with an optional relay mode enabled by a dedicated action). Additionally, imaging requires camera calibration, and sample analysis requires warming up the lab, creating structured precondition dependencies that are violated under hazard-style misconceptions and detoured under cost-based skill misconceptions.

The expert plan is as follows:

```
(calibrate_camera r1 cam1 cal_obj w2)
(enable_relay)
(warm_up_lab r1 w2)
(sample_soil_direct r1 s2 soil1 w2)
(navigate r1 w2 w1)
(establish_link r1 l1 w1)
(communicate_soil r1 l1 w1 soil1)
(navigate r1 w1 w4)
(mark_communicated_soil soil1)
(advance_to_next_item soil1 rock1)
(sample_rock_direct r1 s1 rock1 w4)
(release_channel w1 l1)
(navigate r1 w4 w1)
(establish_link r1 l1 w1)
(communicate_rock r1 l1 w1 rock1)
(navigate r1 w1 w4)
(mark_communicated_rock rock1)
(empty_store r1 s2)
(advance_to_next_item rock1 soil2)
(sample_soil_direct r1 s2 soil2 w4)
(release_channel w1 l1)
(navigate r1 w4 w1)
(establish_link r1 l1 w1)
(communicate_soil r1 l1 w1 soil2)
(navigate r1 w1 w4)
(mark_communicated_soil soil2)
(empty_store r1 s1)
(navigate r1 w4 w3)
(advance_to_next_item soil2 rock2)
(sample_rock_direct r1 s1 rock2 w3)
(navigate r1 w3 w4)
(release_channel w1 l1)
(navigate r1 w4 w1)
(establish_link r1 l1 w1)
(communicate_rock r1 l1 w1 rock2)
(navigate r1 w1 w4)
(mark_communicated_rock rock2)
(advance_to_next_item rock2 obja)
(take_color_image_std r1 cam1 obja cal_obj w4)
```

```
(release_channel w1 l1)
(navigate r1 w4 w1)
(establish_link r1 l1 w1)
(communicate_color_image r1 l1 w1 obja)
(mark_communicated_color_image obja)
(navigate r1 w1 w2)
(navigate r1 w2 w3)
(advance_to_next_item obja objb)
(take_high_image_std r1 cam1 objb cal_obj w3)
(release_channel w1 l1)
(navigate r1 w3 w2)
(establish_link r1 l1 w2)
(communicate_high_image r1 l1 w2 objb)
(mark_communicated_high_image objb)
(end_task)
```

*Listing 3.* Rover expert plan

Below are the descriptions of 13 unique misconceptions simulated for the Overcooked task, categorized as *hazard*, *skill*, or *type* (see Sec. 5.1):

1. Camera Calibration (*Hazard*): The student is unaware that the camera must be calibrated before use, so it skips camera calibration and takes uncalibrated images.

2. Channel Busy (*Hazard*): The student is missing knowledge related to the "channel busy" constraint, typically causing it to ignore channel conflicts or avoid direct links in favor of relays.

3. Establish Link (*Hazard*): The student believes "establish link" does not consume the channel (infinite capacity), so it repeatedly establishes links without releasing them.

4. Lab Warmup (*Hazard*): "The student is unaware that the lab must be warmed up before analysis, so it skips "warmup lab" and attempt raw processing.

5. Clear Channel (*Hazard*): "The student forgets that "release channel" clears the established link, leading to the channel being used when it is already occupied.

6. Empty Storage (*Hazard*): The student fails to realize that it must empty the store before taking new samples, leading it to 'mix' objects in the store rather than clearing them.

7. Costly Soil Communicate (*Skill*): The student believes "communicate soil" is very costly, so it chooses to "relay soil" instead.

8. Costly Soil Relay (*Skill*): The student believes "relay soil" is very costly, so it forces itself to use the direct "communicate soil" action.

9. Costly Release Channel (*Skill*): The student believes "release channel" is expensive, causing it to avoid releasing the channel whenever possible.

10. Costly Rock Sample (*Skill*): The student believes sampling rock directly is expensive, so it uses the alternative raw sampling method.

11. Costly Soil Sample (*Skill*): The student believes sampling soil directly is expensive, so it uses the alternative raw sampling method.

12. Costly Color Image (*Skill*): The student believes taking standard color images is expensive, so it captures uncalibrated/raw images instead.

13. Costly High-Res Image (*Skill*): The student believes taking standard high-res images is expensive, so it captures uncalibrated/raw images instead.

### A.2. Summary of Dataset

The number of knowledge components (# Components), the number of components with simulated gaps (# Gap Components), and the number of unique student plans for 1, 2, and 3 misconceptions (1-Misconception, 2-Misconception, 3-Misconception, respectively) are summarized in Table 3. The 1-Misconception data are used for training, and all other data are used purely for evaluation.

| | 1-Misconception | 2-Misconception | 3-Misconception | # Components | # Gap Components |
|---|---|---|---|---|---|
| Breakfast | 550 | 2646 | 0 | 26 | 13 |
| Overcooked | 168 | 552 | 0 | 51 | 7 |
| Rover | 13 | 50 | 43 | 34 | 17 |

*Table 3.* Summary of each dataset

### A.3. Sample Student Plans and PDDL Snippets

Below is an example of a student with Salt-Sugar Confusion (*Type* misconception) and No Spice Verify Skill (*Skill* misconception), simulated in the Breakfast task. In the student PDDL, the object types of `sugar1` and `salt1` are swapped in the student's `objects` block, indicating type confusion. To simulate the missing spice-verification skill, we assign a prohibitively high cost to the `verify_is_salt` and `verify_is_sugar` actions in the student PDDL. Since the planner minimizes total cost, it avoids these verification actions and generates a plan as if the student does not check the identity of the spices. As a result, the student may season the egg with `sugar1` instead of `salt1`, without first verifying which seasoning is which.

```
### Expert PDDL Snippet ###
(:objects
    raw_egg1 - raw_egg
    boiled_egg1 - boiled_egg
    tomatosauce1 - tomatosauce
    milk1 - spoiled_milk
    milk2 - fresh_milk
    bowl1 bowl2 - bowl
    plate1 - plate
    pan1 - pan
    cup1 - cup
    sugar1 - sugar
    salt1 - salt
    sponge1 metal_brush1 - cleaner
)
(:action verify_is_salt
    :parameters (?s - seasoning)
    :precondition (and
        (is_cooking)
        (is_salt ?s)
    )
    :effect (and
        (verified_salt ?s)
    )
)
(:action verify_is_sugar
    :parameters (?s - seasoning)
    :precondition (and
        (is_cooking)
        (is_sugar ?s)
    )
    :effect (and
        (verified_sugar ?s)
    )
)
```

```
### Student PDDL Snippet ###
(:objects
    raw_egg1 - raw_egg
    boiled_egg1 - boiled_egg
    tomatosauce1 - tomatosauce
    milk1 - spoiled_milk
    milk2 - fresh_milk
    bowl1 bowl2 - bowl
    plate1 - plate
    pan1 - pan
    cup1 - cup
    sugar1 - salt # <- Does not know which item is salt or sugar
    salt1 - sugar
    sponge1 metal_brush1 - cleaner
)
(:action verify_is_salt
    :parameters (?s - seasoning)
    :precondition (and
        (is_cooking)
        (is_salt ?s)
    )
    :effect (and
        (verified_salt ?s)
        (increase (total-cost) 10000) # <- Prohibitive action cost
    )
)
(:action verify_is_sugar
    :parameters (?s - seasoning)
    :precondition (and
        (is_cooking)
        (is_sugar ?s)
    )
    :effect (and
        (verified_sugar ?s)
        (increase (total-cost) 10000) # <- Prohibitive action cost
    )
)
```

*Listing 4.* Breakfast task combined *Type* and *Skill* misconception PDDL example

```
(verify_is_raw_egg raw_egg1)
(serve_milk_fresh milk2 cup1)
(crack_egg_raw raw_egg1 bowl2)
(beat_egg raw_egg1 bowl2)
(season_egg_with_salt raw_egg1 sugar1 bowl2) # <- Student seasons egg with sugar
(preheat_pan pan1)
(transfer_ingredient raw_egg1 bowl2 pan1)
(cook_ingredient_in_preheated raw_egg1 pan1)
(plate_egg raw_egg1 pan1 plate1)
(open_can tomatosauce1)
(dump_can_to tomatosauce1 bowl1)
(microwave_ingredient_safe tomatosauce1 bowl1)
(add_topping raw_egg1 tomatosauce1 bowl1)
(clean_container_soft pan1 sponge1)
(clean_container_harsh_damaging bowl2 metal_brush1)
(clean_container_harsh_damaging bowl1 metal_brush1)
```

*Listing 5.* Breakfast task combined *Type* and *Skill* misconception student plan example

## B. SENSEI Implementation and Training Details

### B.1. Knowledge and Behavior Encoder ($f_{\text{enc}}$)

We directly use a pretrained CodeT5+ encoder as our knowledge and behavior encoder without finetuning.

## B.2. Localization Network ($f_{\text{loc}}$)

The localization network is a pre-layer-normalized, three-layer MLP. The concatenated inputs $(z^E, z^S, \Delta z, e_i^E, e_G)$ are first normalized with layer norm (Ba et al., 2016), then passed, followed by two hidden linear layers with GELU (Hendrycks & Gimpel, 2016) activations and 0.1 dropout. The final linear layer produces a scalar output $\hat{g}_i$. The hidden layer dimension is 512.

In the gap localization BCE loss ($\mathcal{L}_{\text{loc}}$), positive samples are defined as samples with identifiable knowledge gaps, and all other samples are considered negative samples. $\mathcal{L}_{\text{loc}}$ is designed to give a 3 times larger weight to positive samples than to negative samples to compensate for the imbalanced number of components with and without gaps. The weight of each gap component is also adjusted according to the number of identifiable time steps to balance the contributions from different misconceptions.

For the mixup generalization loss ($\mathcal{L}_{\text{mix}}$), "psuedo-multi-misconception" embedding $z^{\text{mix}}$ is generated by linearly interpolating between source embeddings $z^A$ and $z^B$ as $z^{\text{mix}} = \alpha z^A + (1 - \alpha z^B)$, where $\alpha \sim U[0, 1]$.

The two losses are weighted equally ($\lambda_{\text{mix}} = 1$), and the network is trained with a learning rate of 0.0001.

## B.3. Latent Knowledge Editor Network ($f_{\text{edit}}$)

The latent knowledge editor network is a two-layer MLP with pre-layer normalization. The concatenated inputs $(z^E, z^S, \Delta z, e_i^E, e_G)$ is first normalized using layer norm, then passed through a linear projection followed by a GELU nonlinearity. Dropout 0.1 is applied to the hidden representation, and a final linear layer maps the hidden features to the output embedding space. The hidden layer dimension is 256, and the network weights are zero-initialized.

Both editor network losses are weighted equally ($\lambda_{\text{no-op}} = 1$), and the network is training with a learning rate of 0.0001

## B.4. Student Knowledge Decoder Network ($f_{\text{dec}}$)

We initialize the decoder from a pretrained CodeT5+ model, and prepend a learnable projection layer (with pre-layer-norm and tanh activation) to adapt the input representation. The decoder and the projection layer are first pretrained jointly on our dataset. Then the decoder is frozen, and only the projection layer is finetuned together with the remaining model components. Each component is trained with a learning rate of 0.0005.

# C. Additional Experiments

## C.1. Impact of Detection Sensitivity on Recall–Precision Trade-Off

In our primary evaluation, we defined a knowledge gap as detected if the localization network predicted a probability $> 0.5$ for a single timestep. This standard classification setting maximizes the $\text{Sys}_{\text{rec}}$ for SENSEI, ensuring the system identifies as many knowledge gaps as possible. However, while high recall is crucial for safety-critical domains where missing errors are risky, precision may be a higher priority in settings like human-AI collaboration (where minimizing erroneous corrections is essential for maintaining user trust) or time-critical settings (where users cannot afford to go through many erroneous guidance).

To address this, we perform a grid search over two hyperparameters to analyze the impact of varying the guidance guidelines:

- **Detection Threshold** ($p_{\text{thresh}}$): defines the minimum probability required to flag a gap (swept from 0.1 to 0.95). The second parameter, temporal consistency ($k$), is the number of consecutive time steps a gap must be detected before providing guidance (swept from 1 to 5).

- **Temporal Consistency** ($k$): the number of consecutive time steps a gap must be detected before providing guidance (swept from 1 to 5).

Fig. 4 illustrates the system-level precision-recall trade-off for the Breakfast, Overcooked, and Rover tasks. We observe that increasing each parameter increases the precision. While our chosen setting ($p_{\text{thresh}} = 0.5, k = 1$) achieves highest recall, this comes at the cost of lower precision. To achieve high precision ($> 80\%$), which effectively minimizes erroneous corrections, the system operates at a conservative recall of approximately $60\%$. This trade-off highlights the tunability of SENSEI to both high-recall and high-precision settings.

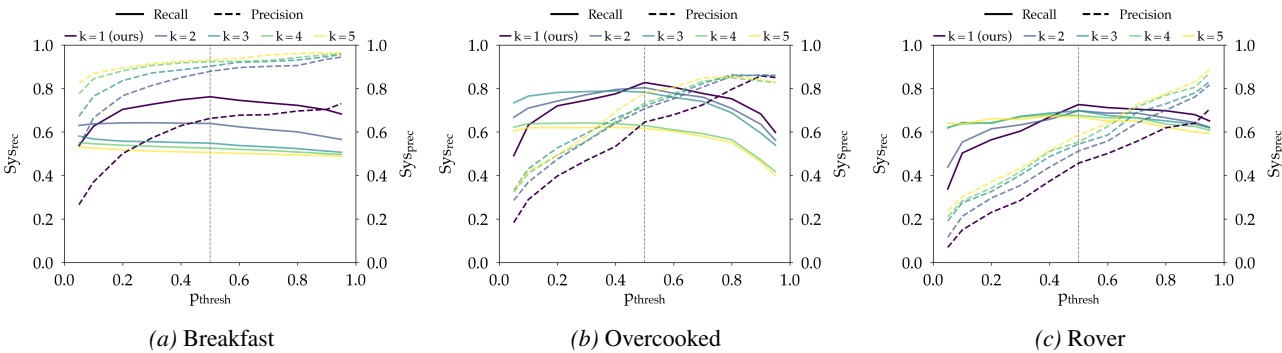

*(a)* Breakfast          *(b)* Overcooked          *(c)* Rover

*Figure 4.* System Recall-Precision Trade-off. Our chosen setting that maximizes system recall is indicated by the vertical line ($p_{\text{thresh}} = 0.5$) and dark purple graph ($k = 1$).

We further analyze the performance limit of SENSEI and the ablation models by tuning $k$ and $p_{\text{thresh}}$ for the highest F1 score. Results are summarized in Table 4. When optimized for the F1 score, SENSEI outperforms other ablations in the Overcooked and Rover tasks. While No-Expert-Traj and No-Expert ablations achieve a higher F1 score for the Breakfast task, SENSEI achieves the highest score on average across the three tasks, indicating robustness of the SENSEI architecture.

*Table 4.* Performance of maximum F1 score settings for ablation models, reported per domain and as the average of all domains.

|  |  | $k$ | $p_{\text{thresh}}$ | $\text{Sys}_{\text{rec}}(\uparrow)$ | $\text{Sys}_{\text{prec}}(\uparrow)$ | $\text{Sys}_{\text{F1}}$ |
|---|---|---|---|---|---|---|
| Breakfast | No-Mixup | 2 | 0.8 | 0.203 | 0.999 | 0.337 |
|  | No-Expert-Traj | 2 | 0.5 | 0.628 | 0.950 | **0.756** |
|  | No-Expert | 1 | 0.5 | 0.627 | 0.950 | 0.755 |
|  | **SENSEI** | 2 | 0.6 | 0.624 | 0.897 | 0.736 |
| Overcooked | No-Mixup | 1 | 0.95 | 0.643 | 0.878 | 0.742 |
|  | No-Expert-Traj | 1 | 0.5 | 0.746 | 0.714 | 0.730 |
|  | No-Expert | 3 | 0.5 | 0.668 | 0.770 | 0.715 |
|  | **SENSEI** | 2 | 0.6 | 0.760 | 0.807 | **0.783** |
| Rover | No-Mixup | 2 | 0.7 | 0.410 | 0.844 | 0.552 |
|  | No-Expert-Traj | 5 | 0.7 | 0.637 | 0.471 | 0.545 |
|  | No-Expert | 1 | 0.7 | 0.658 | 0.614 | 0.635 |
|  | **SENSEI** | 5 | 0.95 | 0.594 | 0.887 | **0.712** |
| 3 Task Avg | No-Mixup | - | - | 0.419 | 0.907 | 0.573 |
|  | No-Expert-Traj | - | - | 0.670 | 0.712 | 0.690 |
|  | No-Expert | - | - | 0.651 | 0.778 | 0.709 |
|  | **SENSEI** | - | - | 0.659 | 0.863 | **0.747** |

### C.2. Incremental Trajectory Training for Causal Sensitivity

In our primary experiments, we train the model on incremental trajectory prefixes (sub-sequences $\tau_{0:t}$, where $t \in \{1, ..., N\}$) rather than only on complete plans, and designate a gap as "detected" if the localization module identifies a component as a gap at any time step within the trajectory.

This design choice is motivated by the need for causal localization. We hypothesize that training on full plans encourages the model to learn spurious correlations between the final state and the error label rather than understanding the procedural deviation. By training on increments, we provide dense supervision, forcing the model to distinguish exactly *when* a trajectory transitions from optimal to suboptimal. This teaches the model to associate specific cues in the plan to the *cause* of the error, leading to stronger compositional generalization capabilities.

To validate this hypothesis, we train a *Full-Plan* variant of SENSEI using as input full trajectory traces $\tau_{0:N}$ only, rather than trajectory increments $\tau_{0:t}$. System performance for the *Full-Plan* model and SENSEI are presented in Table 5.

Empirically, we observe that full-plan training leads to unstable performance: while the full-plan variant performs better than SENSEI for the Rover task in recall (and performs similarly to SENSEI in precision), the system is over-conservative to

identify gaps for Breakfast, and exhibits high erroneous localization in Overcooked. In contrast, our incremental trajectory training strategy consistently balances recall and precision, supporting that localizing and correcting errors as they unfold is essential for robust error identification.

*Table 5.* Effect of Incremental Trajectory Training

|  |  | $\text{Sys}_\text{rec}(\uparrow)$ | $\text{Sys}_\text{prec}(\uparrow)$ | $\text{Sys}_\text{F1}$ |
|---|---|---|---|---|
| Breakfast | Full-Plan | 0.471 | 0.928 | 0.625 |
|  | **SENSEI** | 0.762 | 0.663 | 0.709 |
| Overcooked | Full-Plan | 0.606 | 0.297 | 0.399 |
|  | **SENSEI** | 0.828 | 0.646 | 0.726 |
| Rover | Full-Plan | 0.884 | 0.456 | 0.602 |
|  | **SENSEI** | 0.727 | 0.457 | 0.561 |

## C.3. Effect of Removing $\Delta z$ from Model Inputs

In addition to the individual behavior embeddings of the expert and the student ($z^E$, $z^S$), we provide their different ($\Delta z$) as inputs to the localization and editing modules. The rationale for this design choice was to inject an inductive bias encouraging the model to focus on behavioral "differences." However, explicitly providing $\Delta z$ is theoretically not necessary as it is collinear to $z^E$ and $z^S$. The effect of removing $\Delta z$ is evaluated (*No-$\Delta z$*), and the results are shown in Table 6. The results of other ablation experiments are repeated for reference. Removing $\Delta z$ resulted in mixed but encouraging outcomes. Compared to SENSEI, while the system recall decreased for the Breakfast and Overcooked tasks, precision increased for the Breakfast and Rover tasks, resulting in improved F1 scores.

*Table 6.* Effect of Removing $\Delta z$ from Model Inputs

|  |  | $\text{Sys}_\text{rec}(\uparrow)$ | $\text{Sys}_\text{prec}(\uparrow)$ | $\text{Sys}_\text{F1}(\uparrow)$ |
|---|---|---|---|---|
| Breakfast | No-Mixup | 0.221 | 0.954 | 0.359 |
|  | No-Expert-Traj | 0.673 | 0.793 | 0.728 |
|  | No-Expert | 0.635 | 0.923 | 0.752 |
|  | No-$\Delta z$ | 0.688 | 0.848 | **0.759** |
|  | **SENSEI** | **0.762** | 0.663 | 0.709 |
| Overcooked | No-Mixup | 0.690 | 0.707 | 0.698 |
|  | No-Expert-Traj | 0.774 | 0.670 | 0.718 |
|  | No-Expert | 0.747 | 0.672 | 0.708 |
|  | No-$\Delta z$ | 0.788 | 0.637 | 0.704 |
|  | **SENSEI** | **0.828** | 0.646 | **0.726** |
| Rover | No-Mixup | 0.457 | 0.743 | 0.566 |
|  | No-Expert-Traj | 0.673 | 0.365 | 0.473 |
|  | No-Expert | 0.680 | 0.512 | 0.584 |
|  | No-$\Delta z$ | **0.815** | 0.543 | **0.652** |
|  | **SENSEI** | 0.727 | 0.457 | 0.561 |

## C.4. Random Baseline with Adjusted Detection Threshold

The Random-Loc baseline evaluated in the main paper declares each knowledge component as a gap with 50% probability at each plan time step. We consider a knowledge component to be labeled as a gap if it was declared as a gap at any point in the plan. Since the Random-Loc baseline labels most components as a gap at some point in the plan, it results in near perfect localization recall and low localization precision. Here, we evaluate an alternative random localization baseline (Random-Avg) that uses a gap identification probability such that the expected number of gaps detected matches the average number of gaps seen in students in the training set. Results are shown in Table 7.

As expected, the Random-Avg setting results in significantly lower $\text{Loc}_\text{rec}$ and $\text{Sys}_\text{false}$, and limited effect on $\text{Loc}_\text{prec}$. The F1 scores decrease for all tasks when compared to Random-Loc, further supporting that localization is not a trivial task.

*Table 7.* Random localization baseline with adjusted gap identification probability

| | | $Loc_{rec}(\uparrow)$ | $Loc_{prec}(\uparrow)$ | $Loc_{F1}(\uparrow)$ | $Edit_{acc}(\uparrow)$ | $Edit_{no\text{-}op}(\uparrow)$ | $Sys_{rec}(\uparrow)$ | $Sys_{prec}(\uparrow)$ | $Sys_{F1}(\uparrow)$ | $Sys_{false}(\downarrow)$ |
|---|---|---|---|---|---|---|---|---|---|---|
| Breakfast | Random-Loc | 0.997 | 0.104 | 0.188 | 0.875 | 0.0 | 0.875 | 0.092 | 0.166 | 0.999 |
| | Random-Avg | 0.040 | 0.093 | 0.056 | 0.920 | 0.0 | 0.037 | 0.085 | 0.052 | 0.046 |
| Overcooked | Random-Loc | 1.0 | 0.040 | 0.077 | 0.925 | 0.0 | 0.925 | 0.036 | 0.069 | 1.0 |
| | Random-Avg | 0.026 | 0.053 | 0.035 | 0.930 | 0.0 | 0.024 | 0.049 | 0.032 | 0.019 |
| Rover | Random-Loc | 1.0 | 0.088 | 0.162 | 0.940 | 0.040 | 0.939 | 0.083 | 0.153 | 0.957 |
| | Random-Avg | 0.048 | 0.083 | 0.061 | 1.0 | 0.03 | 0.048 | 0.083 | 0.061 | 0.049 |

# D. User Study Details

20 users (3 females, 17 males), ages 20 to 30, participated in the study.

## D.1. Modified Overcooked AI Environment

The original environment (typically for multi-agent collaboration) was modified for a single-player setting, and the user control inputs were changed from keyboard input to a point-and-click UI. To support a wider range of misconceptions, we additionally implemented "doors" that unlock shortcut paths to items when opened, as well as a distractor white counter object (the same color as the serving location, but it is a regular counter) and a hidden serving location (same color as regular counters). Finally, we implemented a "careful" version of the "pick up" action that must be used when picking up items that must be handled with care (a list of items was provided to the users). The Overcooked environment used in the user study is shown in Fig. 5. The user's actions in the environment (e.g., move to pot, cook soup, open door to onion, etc.) were converted to a PDDL-solution-style symbolic plan to feed into SENSEI for localization and correction.

## D.2. Induced Misconceptions

All users were given the same tutorial material, which went through how to open unlocked doors, pick up items ("carefully" and regularly), place items into the pot, cook the soup, plate the soup, and serve the soup.

For each group, we induced 2 misconceptions by either withholding certain information or by modifying the environment. The following misconceptions were used:

- **No Open Onion Door**: Door to the onions are locked via a change in the environment. The door is openable by pressing the "A" key in front of the doors, but users are not told how to open the doors.

- **No Open Tomato Door**: Same as above, but for the door to the tomatoes.

- **No Open Dish Door**: Same as above, but for the door to the dishes.

- **Careless Tomato**: Tomatoes require "careful" pickup, but this information was not given to the users.

- **Careless Dish**: Dishes require "careful" pickup, but this information was not given to the users.

- **Counter Confusion**: The serving location was hidden by making it the same color as surrounding counters. One of the standard colors were changed to the serving location color to serve as a distractor.

## D.3. User Study Procedure

Users were first asked to complete a pre-study survey, followed by a step-by-step tutorial task to familiarize with the interface. The users then completed the first trial (Study 1) with 2 misconceptions, and SENSEI printed targeted guidance to a terminal. The users completed the second trial (Study 2) after reading SENSEI's guidance, then concluded with a post-study survey.

Natural language guidance was generated by using a predefined mapping from a predicted (localized and corrected) PDDL block to a guidance text. When the system predicted one of the misconceptions from Section D.2, a corresponding guidance message is printed out to the terminal. If a predicted misconception was not one of the expected misconceptions defined in the mapping, the system outputs a placeholder guidance saying "A knowledge gap was detected, but there are no additional information that can be disclosed." The guidance text corresponding to each misconception is listed below:

- **No Open Onion Door**: "If the DOOR to the ONION does not open by clicking, it opens by pressing the 'a' key in front of the door."

- **No Open Tomato Door**: "If the DOOR to the TOMATO does not open by clicking, it opens by pressing the 'a' key in front of the door."

- **No Open Dish Door**: "If the DOOR to the DISH does not open by clicking, it opens by pressing the 'a' key in front of the door."

- **Careless Tomato**: "A TOMATO should be handled with care. Use the CAREFUL ACTION to pick up the tomato."

- **Careless Dish**: "A DISH should be handled with care. Use the CAREFUL ACTION to pick up the dish."

- **Counter Confusion**: "The SERVING COUNTER is the lower-most grid on the right-most column."

### D.4. Per-Group Quantitative Results

Misconception assignments to each group in the user study are as follows: G1 (No Open Door Tomato, No Open Door Dish), G2 (No Open Door Dish, Careless Tomato), G3 (Careless Tomato, Careless Dish), G4 (No Open Door Onion, Counter Confusion). Per-group system recall ($Sys_{rec}$), system precision ($Sys_{prec}$), and plan alignment scores $IoU_{pre}$ (before guidance) and $IoU_{post}$ (after guidance) are shown in Table 8.

*Table 8.* Per-Group User Study Results. Each row contains the mean and variance of metrics across 5 subjects for each group (G). The final row are mean and variance across all 20 subjects.

|     | $Sys_{rec}$ | $Sys_{prec}$ | $IoU_{pre}$ | $IoU_{post}$ |
| --- | --- | --- | --- | --- |
| G1 | $0.900 \pm 0.050$ | $0.533 \pm 0.089$ | $0.711 \pm 0.010$ | $0.806 \pm 0.001$ |
| G2 | $0.800 \pm 0.075$ | $0.466 \pm 0.019$ | $0.759 \pm 0.003$ | $0.824 \pm 0.010$ |
| G3 | $1.000 \pm 0.000$ | $0.600 \pm 0.022$ | $0.805 \pm 0.004$ | $0.932 \pm 0.011$ |
| G4 | $0.900 \pm 0.050$ | $0.640 \pm 0.123$ | $0.666 \pm 0.002$ | $0.819 \pm 0.004$ |
| All | $0.895 \pm 0.042$ | $0.555 \pm 0.058$ | $0.736 \pm 0.007$ | $0.844 \pm 0.008$ |

### D.5. Post-Study Survey

Users were asked to complete a post-study survey following their participation in the experiment. The following questions were asked:

Q1. How many of the additional tips did you find helpful?

Q2. How many of the additional tips did you NOT find helpful?

Q3. Overall, how helpful were the additional tips in providing you with information you did not know? (Likert scale from 1-5 where 1 is "not helpful at all" and 5 is "extremely helpful")

Q4. Overall, how annoying were the additional tips? Please rate the content, disregarding wording and tone. (Likert scale from 1-5 where 1 is "not annoying at all" and 5 is "very annoying")

Post-study survey responses averaged across 20 users are summarized in Table 9.

## E. Off-the-Shelf LLM Baseline Details

Here we provide additional details on the off-the-shelf LLM baselines. First, each LLM model is prompted with the system prompt shown in Listing 6 to identify PDDL blocks where knowledge gaps exist, and output a rewritten PDDL block and a natural language description of the student misconception for each identified gap block. Each output student PDDL block were extracted, and all blocks correctly identified as gaps were evaluated according to three different criteria of varying flexibility:

*Table 9.* Post-Study Survey Results

| Question | Response |
|:---:|:---:|
| Q1 | $1.91 \pm 0.087$ |
| Q2 | $1.64 \pm 2.340$ |
| Q3 | $4.27 \pm 0.589$ |
| Q4 | $2.27 \pm 1.160$ |

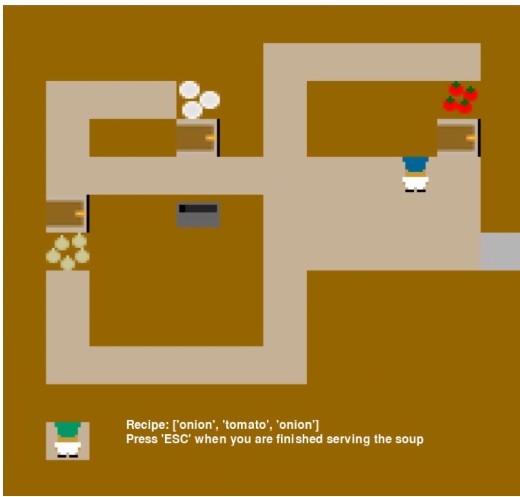

*Figure 5.* User study environment. Users can click on items to move to and interact with items in the scene. The recipe is displayed at the bottom of the screen.

- **Semantic Equivalence** (most strict): Two PDDL blocks are considered equivalent if they are semantically identical (invariant to spacing or ordering irrelevant to the definition of the block). This is the criteria used to evaluate all methods in the main paper.

- **Conceptual Equivalence** (middle ground): Two PDDL blocks are considered equivalent if they express the same effects. For example, the LLM adds an extra `(no_need_careful)` predicate to the action precondition and removes a negative effect `(harmless_pickup)`, where the ground truth only removes the negative effect. While this edit results in an invalid PDDL (by using undefined predicated), the conceptual meaning interpreted from the two PDDL blocks are equivalent.

- **Explanation Equivalence** (most lenient): This criteria compares the LLM-generated description of the student misconception in the edited PDDL block to a set of ground truth misconception descriptions in Appendix A.1.

GPT-4o was prompted with each of the evaluation prompts (Listings 7, 8, 9) to measure correction success of the LLM baselines under the three criteria. Results are shown in Table 10. Note that the system metrics are conditioned on the localization accuracy (Loc$_{rec}$), which is independent of the evaluation criteria presented here. As described in the main results (Table 1), the primary performance bottleneck for these baselines is Loc$_{rec}$. However, we adopt this evaluation metric to evaluate the LLM baselines for the same task as other methods (localization and correction).

*Table 10.* Off-the-Shelf LLM Baseline Results for Semantic, Concept, and Explanation Metrics

| | | | $\text{Sys}_{\text{rec}}(\uparrow)$ | $\text{Sys}_{\text{prec}}(\uparrow)$ | $\text{Sys}_{F1}(\uparrow)$ |
|---|---|---|---|---|---|
| GPT-4o | Breakfast | Semantic | 0.077 | 0.177 | 0.107 |
| | | Concept | 0.084 | 0.194 | 0.117 |
| | | Explanation | 0.1015 | 0.242 | 0.144 |
| | Overcooked | Semantic | 0.005 | 0.005 | 0.005 |
| | | Concept | 0.010 | 0.011 | 0.010 |
| | | Explanation | 0.100 | 0.106 | 0.103 |
| | Rover | Semantic | 0.0 | 0.0 | 0.0 |
| | | Concept | 0.0 | 0.0 | 0.0 |
| | | Explanation | 0.008 | 0.016 | 0.011 |
| GPT-5.2 | Breakfast | Semantic | 0.101 | 0.166 | 0.126 |
| | | Concept | 0.101 | 0.166 | 0.126 |
| | | Explanation | 0.126 | 0.206 | 0.156 |
| | Overcooked | Semantic | 0.085 | 0.069 | 0.076 |
| | | Concept | 0.085 | 0.069 | 0.076 |
| | | Explanation | 0.215 | 0.175 | 0.193 |
| | Rover | Semantic | 0.0 | 0.0 | 0.0 |
| | | Concept | 0.004 | 0.009 | 0.006 |
| | | Explanation | 0.016 | 0.034 | 0.022 |
| Llama-3.3 70B | Breakfast | Semantic | 0.007 | 0.013 | 0.009 |
| | | Concept | 0.024 | 0.044 | 0.031 |
| | | Explanation | 0.090 | 0.163 | 0.116 |
| | Overcooked | Semantic | 0.035 | 0.019 | 0.025 |
| | | Concept | 0.145 | 0.078 | 0.101 |
| | | Explanation | 0.095 | 0.051 | 0.066 |
| | Rover | Semantic | 0.0 | 0.0 | 0.0 |
| | | Concept | 0.008 | 0.014 | 0.010 |
| | | Explanation | 0.012 | 0.021 | 0.0153 |

```
SYSTEM: You are a Cognitive Scientist and PDDL Expert analyzing a student's behavior.

CONTEXT:
You have the "Expert Domain and Problem" (Ground Truth Physics), an "Expert Plan" (optimal
    behavior), and a "Student Plan" (trace of actions).
The student has a FLAWED mental model of the domain. They believe their plan is optimal,
    but it is based on FALSE BELIEFS about the world.
Crucially, the student's actions are **valid PDDL transitions** in their mind, but they
    may be invalid or suboptimal in the real world.
The "Expert Plan" is an example of an optimal plan, and is not the only optimal plan.

The student's error usually falls into one of these categories:
1. Incorrect Action Logic (e.g., wrong effects).
2. Incorrect Object Definitions (e.g., they think 'onion1' is type 'tomato').

INPUTS:
[EXPERT DOMAIN AND PROBLEM PDDL]
{expert_pddl}

[EXPERT PLAN]
{expert_plan}

[STUDENT PLAN]
{student_plan}

TASK:
1. Trace the plan step-by-step and compare it to optimal behavior (expert plan or equally
    good plan).
2. Identify the FIRST step where the student's behavior diverges from what a rational
```

```
      expert would do.
3. INFER the specific knowledge gap(s). Note that a single behavioral error might imply
    multiple PDDL changes (e.g., adding an object AND changing a type).
4. For EACH inferred gap, identify the specific PDDL block (e.g., Action, Objects) and
    REWRITE it exactly as the student believes it to be.

IMPORTANT CONSTRAINTS:
- For Action definitions: Output the FULL `(:action ...)` block.
- For Objects blocks: Output the FULL `(:objects ...)` block containing ALL items, not
    just the changed ones, so it can be parsed as valid PDDL.
- Step Index: Use 0-based indexing for the plan steps.

OUTPUT FORMAT:
Return a single JSON object containing a list of gaps under the key "identified_gaps".
Example structure:
{{
  "divergence_step_index": 5,
  "identified_gaps": [
    {{
      "component_type": "action",
      "component_name": "pick-up",
      "explanation": "Student thinks they don't need to hold the knife.",
      "student_pddl_block": "(:action pick-up ... (valid pddl) ...)"
    }},
    {{
      "component_type": "objects",
      "component_name": "problem_objects",
      "explanation": "Student believes onion1 is a tomato.",
      "student_pddl_block": "(:objects onion1 - tomato ...)"
    }}
  ]
}}
```

*Listing 6.* Full System Prompt for In-Context Learning Baselines

```
SYSTEM: You are a PDDL Expert and Strict Logician.
You are evaluating whether a "Predicted PDDL Block" is SEMANTICALLY EQUIVALENT to the "
    Ground Truth Block".

INPUTS:
[GROUND TRUTH PDDL]
{true_student_chunk}

[PREDICTED PDDL]
{llm_predicted_chunk}

TASK:
Determine if the Prediction implements the **exact same logic** as the Ground Truth.
- IGNORE: Whitespace, ordering of predicates, parameter names (e.g., ?p vs ?player), or
    minor syntax typos.
- FOCUS: Are the same preconditions required? Are the same effects applied?
- CRITICAL: If the Prediction adds a predicate that does not exist in the Ground Truth, or
     misses one, it is NOT a match.

OUTPUT:
Return JSON:
{{
  "reasoning": "Step-by-step comparison of logic...",
  "is_semantically_equivalent": boolean
}}
```

*Listing 7.* Full System Prompt for In-Context Learning Baseline Evaluation (Semantic)

```
SYSTEM: You are a Cognitive Scientist analyzing models of student behavior.
```

```
Your goal is to determine if a "Predicted PDDL Update" reflects the **same underlying
    belief** as the "Ground Truth PDDL Update", even if the implementation is different or
     invalid.

CONTEXT:
We are modeling a student who has a false belief about the world.
- The **Ground Truth** is the canonical PDDL representation of this false belief.
- The **Prediction** is an AI's attempt to represent this same belief.

INPUTS:
[GROUND TRUTH PDDL BLOCK]
{true_student_chunk}

[PREDICTED PDDL BLOCK]
{llm_predicted_chunk}

TASK:
Decide if the **agent knowledge inferred** from the Prediction is effectively the same as
    the Ground Truth.
Ask yourself: "If a student believed the Prediction, would they behave effectively the
    same way as a student who believed the Ground Truth?"

GUIDELINES for "MATCH" (True):
1. **Synonyms/Paraphrasing:** GT uses `(hand-empty)` but Prediction uses `(not (holding ?x
    ))`. -> MATCH.
2. **Hallucinated Predicates:** GT uses `(reachable ?x)` but Prediction uses `(can-grab ?x
    )` (which doesn't exist). The intent is clearly the same. -> MATCH.
3. **Implicit vs Explicit:** GT deletes a precondition. Prediction keeps the precondition
    but adds an effect that trivially satisfies it. -> MATCH.

GUIDELINES for "MISMATCH" (False):
1. **Different Logic:** GT says "Student thinks they can't fail," Prediction says "Student
     thinks they succeed instantly." (Subtle, but maybe different).
2. **Wrong Object:** GT refers to `onion1`. Prediction refers to `tomato1`. -> MISMATCH.
3. **Opposite Effect:** GT negates a predicate. Prediction makes it positive. -> MISMATCH.

OUTPUT:
Return JSON:
{{
  "inferred_belief_ground_truth": "One sentence summary of what the GT implies (e.g., '
     Agent believes holding isn't required').",
  "inferred_belief_prediction": "One sentence summary of what the Prediction implies.",
  "is_conceptually_equivalent": boolean
}}
```

*Listing 8.* Full System Prompt for In-Context Learning Baseline Evaluation (Concept)

```
SYSTEM: You are a Cognitive Scientist evaluating student diagnosis.
You have a "Ground Truth Diagnosis" (the actual misconception) and a "Predicted Diagnosis"
     (generated by an AI tutor).

INPUTS:
[GROUND TRUTH EXPLANATION]
{gt_explanation}

[PREDICTED EXPLANATION]
{llm_explanation}

TASK:
Do these two explanations describe the **same underlying knowledge gap**?
- TRUE: "Student thinks onion is tomato" vs "Student confuses onion object with tomato
    type".
- FALSE: "Student thinks onion is tomato" vs "Student thinks they don't need an onion".

OUTPUT:
```

```
Return JSON:
{{
  "reasoning": "Compare the core concept...",
  "is_same_knowledge_gap": boolean
}}
```

*Listing 9.* Full System Prompt for In-Context Learning Baseline Evaluation (Explanation)

