# OpenReview forum: "Fix the Mind, Not the Move: Interpretable AI Assistance via Knowledge-Gap Localization"
_ICML.cc/2026/Conference — ICML 2026 regular_

### Official Review · Reviewer_BLzX · 2026-03-08

**Soundness:** 3
**Presentation:** 4
**Significance:** 3
**Originality:** 3
**Overall Recommendation:** 5
**Confidence:** 4

**Summary:**

The authors present a novel architecture for an AI assistant. Their framing focuses on filling in user knowledge gaps instead of simply suggesting a corrected behavior. The authors test their architecture in 3 simulated environments, performing comparisons to LLM baselines and ablation studies by modifying components of their architecture. They also perform a user survey where they test if their system successfully identifies and corrects the participants' knowledge gaps.

**Compliance With Llm Reviewing Policy:**

Affirmed.

**Final Justification:**

The rebuttal addressed my concerns and I have raised my score.

**Key Questions For Authors:**

1. Shouldn't the final output of the pipeline in Fig. 2 be $\hat{v}_i^S$ ? The hat is missing.

2. Eq. (4): Is $\Delta z$ needed? It is collinear to $z^E, z^S$ and the first layer weights can be trivially updated to produce the same output.

3. Do you commit on providing a public implementation of your method and experiments?

**Limitations:**

yes

**Strengths And Weaknesses:**

Strengths:

* The paper is well written, describing all parts of their architecture and training method in sufficient detail. The visualization of the architecture is also easy to read.
* Framing the expert and student knowledge as a set of interpretable components, and disentangling the different components of the architecture make the system relatively interpretable, as it is easier to detect which components failed and where the model's assessment of the student's knowledge and their true knowledge are misaligned.

Weaknesses:

* The simulated environments are quite far from a real-world scenario and there is no comparison to other methods (although the authors do justify this). Some of the architectural choices are also found to be sub-optimal in terms of F1-score (table 2).

* No implementation is provided with the submission and there are no references in the paper on providing a public implementation upon publication

---

> ### Author Rebuttal · Authors · 2026-03-31
>
> We sincerely thank the reviewer for their constructive comments.
>
> ### W1.1 Real-World Complexity
> Simulating real-world human behavior is a challenge. To rigorously evaluate SENSEI’s diagnostic abilities, we balance complex real-world noise with controlled environments where the “ground truth” misconception is known. Using established benchmarks (Overcooked, IPC Rover) and a custom Breakfast task, we test compositional generalization. Our 3 categories (Type, Hazard, Skill) represent fundamental ways symbolic plans fail (misclassification, ignored negative effects, lack of skill) and each misconception was authored and verified by humans to ensure logical consistency. Though trained only on single misconceptions, SENSEI handles combinatorial complexity zero-shot (e.g., in Breakfast, 11 base misconceptions yield 66 unique, overlapping 2-misconception student profiles). We also inject behavioral variance into the students by generating diverse, valid action permutations for the same underlying knowledge (Sec. 5.1). Further dataset details are in Appendix A.1.
>
> Real-world human errors are complex and include execution failures (e.g., rushing), and our categories are a starting point, not an exhaustive list. However, SENSEI's architecture is not bound by these categories. Since SENSEI decouples knowledge complexity from model complexity, it scales to complex real-world misconceptions by expanding the PDDL representations and collecting human data. Discussion on applying this iterative human-in-the-loop data collection will be added to Sec. 5.1 of the final version.
>
> ### W1.2 Comparison to Existing Methods
> We appreciate the reviewer acknowledging our baseline justification. Because our problem formulation strictly focuses on knowledge alignment (Eq. 3), non-knowledge-based methods without interpretable knowledge predictions trivially fail. For rigorous evaluation, we included a Monolithic End-to-End model representing an alternative neural architecture, and SoTA LLMs representing the strongest off-the-shelf reasoning and text generation approaches.
>
> ### W1.3 Additional Discussion on Ablation Performance
> We view the No-Expert model’s higher F1 in certain domains not as a failure of the main architecture, but as a demonstration of how the anchoring strategy impacts the model's precision-recall trade-off. As shown in Table 2, No-Expert yields lower recall and higher precision than SENSEI. We hypothesize that this is a result of the underlying training mechanism. The No-Expert variant is trained using a random student as the anchor rather than the expert (Sec. 5.6). This forces the model to learn mappings from behavioral differences to knowledge gaps between two suboptimal, noisy agents, making it conservative in declaring knowledge gaps. When evaluated with the expert anchor at inference, this prevents the model from penalizing valid student path variants, reducing false positives. While this improves precision, it harms recall. As No-Expert was trained to require stronger evidence to declare a knowledge gap, it tends to miss subtler behavioral deviations when paired with the expert trajectory. Conversely, SENSEI (expert-anchored during training and inference) is highly sensitive and maintains robust performance across all tasks. When optimized for system F1 (Appendix C.2), SENSEI achieves the highest 3-task average system F1 score (0.761 vs No-Expert’s 0.714). Ultimately, No-Expert’s competitive performance is encouraging, demonstrating our framework’s potential for general agent-agent alignment, even in the absence of a definitive expert. We will add this discussion to Sec. 5.6.
>
> ### W2 Public Implementation
> We commit to a full open-source release upon publication, including: (1) Training/evaluation scripts, user study code, and environments, (2) PDDL data for experts and students (107 unique student misconception combinations), and (3) Optimal expert and ~4000 student behaviors, annotated with ground-truth gaps and plan timesteps where each gap becomes identifiable.
>
> ### Q1 Missing Hat in Fig. 2
> Thank you for catching this. We will correct it in the final version.
>
> ### Q2 Need for $\Delta z$
> As the reviewer points out, $\Delta z$ is collinear to $z^S$ and $z^E$. Our original rationale for explicitly providing $\Delta z$ was to inject an inductive bias encouraging the model to focus on behavioral “differences”. However, prompted by the reviewer’s question, we ran an ablation study removing $\Delta z$ from the input. Removing $\Delta z$ resulted in mixed but encouraging outcomes. While the system recall decreased for all tasks compared to SENSEI, precision increased for the Breakfast and Rover tasks, resulting in improved F1 scores. Here are the $\Delta z$ ablation results (system recall, system precision, system F1): Breakfast (0.688, 0.848, 0.0768), Overcooked (0.788, 0.637, 0.713), Rover (0.815, 0.543, 0.679). We will include this result in our final version.

---

> > ### Author Rebuttal · Reviewer_BLzX · 2026-04-01
> >
> > My concerns have been addressed. I have raised my score to 5 (accept).

---

### Official Review · Reviewer_3X1V · 2026-03-12

**Soundness:** 3
**Presentation:** 3
**Significance:** 3
**Originality:** 3
**Overall Recommendation:** 5
**Confidence:** 3

**Summary:**

The existing AI assistance systems primarily correct users at the action level, directly providing instructions when users make mistakes. However, this paper argues that it is important to identify the underlying misconceptions leading to these mistakes and then address the misconceptions. Based on this idea, the authors propose the SENSEI framework, which formalizes the assistance problem in long-term decision-making tasks as knowledge-gap localization and knowledge editing.

The authors evaluate SENSEI on three complex planning tasks and show that the proposed method outperforms existing methods. Finally, the authors also conduct a user study with 20 users, demonstrating that the proposed method can correct real human misconceptions.

**Compliance With Llm Reviewing Policy:**

Affirmed.

**Final Justification:**

My concerns have been fully addressed, thereby I raise my score.

**Key Questions For Authors:**

How can the proposed method deal with tasks that require continuous actions?

**Limitations:**

Yes

**Strengths And Weaknesses:**

**Strengths:**

The research topic is interesting.

The authors proposed a method to localize the knowledge gap and then edit the knowledge.

The authors verify SENSEI on three complex planning tasks and a real-world user study.

**Weaknesses:**

The method assumes expert knowledge and expert behavior are an oracle. However, in some real-world tasks, there exist several solutions to achieve the same goal.

The method requires that PDDL knowledge components can be explicitly defined and partitioned. However, there exist continuous and fuzzy tasks in the real world.

---

> ### Author Rebuttal · Authors · 2026-03-31
>
> We sincerely thank the reviewer for insightful and constructive comments. Please find our response below.
>
> ### W1 Assumption of a Single Optimal Behavior
> We agree with the reviewer that real-world tasks often have multiple optimal solutions to achieve the same goal. SENSEI inherently handles this multimodality because its primary objective is knowledge alignment, not strict trajectory matching.
>
> Reflecting this objective, our primary system evaluation metrics (in Table 1) are computed strictly in the knowledge space rather than the behavior space, thus measuring the student’s underlying understanding instead of exact behavior match. At the problem formulation level, we formalize this by defining student behavior alignment using post-guidance regret (Eq. 1). Because regret can be defined flexibly (e.g., as performance gap or cost gap), the formulation accommodates any student behavior that is “expert-quality,” regardless of the specific action sequence. Finally, when we evaluate behavior directly in our user study, we utilize intersection over union (IoU) as a metric for behavior alignment. This metric is inherently invariant to the order of actions. If a student completes the task as optimally as the expert but with a different action sequence, the IoU metric correctly recognizes the student behavior as aligned with the expert. While strict order-invariance could theoretically over-reward suboptimal sequences (e.g., taking a long detour and then uselessly opening a shortcut door), we explicitly verified all user trajectories to ensure no such sequence violations occurred in our user study. Thus, our IoU reporting accurately captures the semantic equivalence of valid, multimodal human plans.
>
> ### W2 Reliance on PDDL Representation and Applicability to Continuous Tasks
> We thank the reviewer for their insightful questions regarding the scope of our framework. We would like to clarify that SENSEI does not assume or require knowledge to be modeled with PDDL. As detailed in our formulation (Sec. 3, under “Behavior Alignment” and “Knowledge Components”), the framework only requires two elements: (1) an interpretable knowledge representation that factorizes into semantically meaningful components relevant to the task, and (2) observable agent behaviors that are induced by that knowledge.
>
> We selected PDDL for our empirical evaluation because it is a highly standardized language in the planning community [1, 2, 3, 4, 5], its code block structure naturally serves as “semantically meaningful knowledge components”, and allows us to generate and verify behavior through off-the-shelf solvers. However, as noted in Sec. 4 (under “Knowledge Representation”), our problem formulation allows flexibility in how knowledge is represented. SENSEI could easily be adapted for other interpretable formats, such as talent assessment reports paired with human evaluation behavior, source code defining an algorithmic policy, or interpretable reward functions or cost maps.
>
> Applying this framework to continuous, highly dynamic real-world tasks like autonomous driving is an exciting future direction that SENSEI’s architecture supports. If a continuous agent’s policy is conditioned on an interpretable, decomposable structure (such as a set of safety constraints or parameterized reward functions), our method applies. For example, if a driving agent acts upon a flawed belief about a speed limit or a traffic law constraint, SENSEI could be trained to map the continuous driving behavior back to the specific misconception. We acknowledge that continuous action spaces introduce significantly more behavioral variance and noise, which would require larger datasets and potentially more advanced behavior embedding methods. Nonetheless, the core methodology of mapping behavioral deviations to structured knowledge components scales naturally to these domains.
>
> [1] McDermott et al., PDDL - The Planning Domain Definition Language. AIPS, 1998.
> [2] Long et al., arXiv:1106.5998
> [3] Richter et al., arXiv:1401.3839
> [4] Liu et al., arXiv:2304.11477
> [5] Hao et al., arXiv:2410.12112

---

> > ### Author Rebuttal · Reviewer_3X1V · 2026-04-02
> >
> > The authors fully resolve my concerns.

---

### Official Review · Reviewer_hRsA · 2026-03-13

**Soundness:** 2
**Presentation:** 3
**Significance:** 2
**Originality:** 3
**Overall Recommendation:** 4
**Confidence:** 4

**Summary:**

The paper introduces SENSEI, a framework that tries to fix why people make mistakes rather than just correcting the mistakes themselves. By analyzing a user’s behavior in planning tasks, it identifies the underlying knowledge gaps and provides minimal, targeted guidance to correct them. Experiments and a small user study show that the system can detect misconceptions and improve users’ performance on long-horizon tasks.

**Compliance With Llm Reviewing Policy:**

Affirmed.

**Final Justification:**

I maintain my positive score and decision.

**Key Questions For Authors:**

See weaknesses

**Limitations:**

See weaknesses

**Strengths And Weaknesses:**

Strengths:

1.The perspective of this paper is quite novel: it moves AI assistance from action control or trajectory planning to the level of cognitive diagnosis and education.
2.The model is trained only on single-error data, yet it can handle complex behaviors with multiple intertwined logical errors at inference time through an elegant latent mixup loss, greatly reducing the cost of collecting training data.
3.The figures are clear and the experiments are thorough.

Weaknesses:

The whole framework assumes that the task can be perfectly converted into a PDDL-based state machine, which limits its applicability. As a result, the proposed method may be difficult to apply to highly continuous real-world tasks with fuzzy boundaries, such as strategic interactions in autonomous driving or open-ended creative tasks like painting.

---

> ### Author Rebuttal · Authors · 2026-03-31
>
> We sincerely thank the reviewer for insightful and constructive comments. Please find our response below.
>
> ### W1 Reliance on PDDL Representation and Applicability to Continuous Tasks
> We thank the reviewer for their insightful questions regarding the scope of our framework. We would like to clarify that SENSEI does not assume or require knowledge to be modeled with PDDL. As detailed in our formulation (Sec. 3, under “Behavior Alignment” and “Knowledge Components”), the framework only requires two elements: (1) an interpretable knowledge representation that factorizes into semantically meaningful components relevant to the task, and (2) observable agent behaviors that are induced by that knowledge. We selected PDDL for our empirical evaluation because it is a highly standardized language in the planning community [1, 2, 3, 4, 5], its code block structure naturally serves as “semantically meaningful knowledge components”, and allows us to generate and verify behavior through off-the-shelf solvers. However, as noted in Sec. 4 (under “Knowledge Representation”), our problem formulation allows flexibility in how knowledge is represented. SENSEI could easily be adapted for other interpretable formats, such as talent assessment reports paired with human evaluation behavior, source code defining an algorithmic policy, or interpretable reward functions or cost maps.
>
> Applying this framework to continuous, highly dynamic real-world tasks like autonomous driving is an exciting future direction that SENSEI’s architecture supports. If a continuous agent’s policy is conditioned on an interpretable, decomposable structure (such as a set of safety constraints or parameterized reward functions), our method applies. For example, if a driving agent acts upon a flawed belief about a speed limit or a traffic law constraint, SENSEI could be trained to map the continuous driving behavior back to the specific misconception. We acknowledge that continuous action spaces introduce significantly more behavioral variance and noise, which would require larger datasets and potentially more advanced behavior embedding methods. Nonetheless, the core methodology of mapping behavioral deviations to structured knowledge components scales naturally to these domains.
>
> Regarding creative tasks, we agree with the reviewer that open-ended creative tasks like painting present a distinct challenge. As noted in our discussion (Sec. 6) and Impact Statement, SENSEI inherently adopts a “normative notion of optimality” defined by the expert. Because there often exists no single “optimal” behavior for creative tasks, using SENSEI for instruction in purely creative domains is currently out of scope. However, there exists exciting potential for using SENSEI’s diagnostics for “assisting” users in creative tasks, for instance, by powering adaptive user interfaces that seamlessly surface the right tools and tutorials according to user expertise.
>
> [1] McDermott et al., PDDL - The Planning Domain Definition Language. AIPS, 1998.
> [2] Long et al., arXiv:1106.5998
> [3] Richter et al., arXiv:1401.3839
> [4] Liu et al., arXiv:2304.11477
> [5] Hao et al., arXiv:2410.12112

---

> > ### Author Rebuttal · Reviewer_hRsA · 2026-04-04
> >
> > Thank you for the rebuttal. The authors have largely addressed my concerns, and I will maintain my positive score.

---

### Official Review · Reviewer_VPdY · 2026-03-20

**Soundness:** 2
**Presentation:** 2
**Significance:** 2
**Originality:** 2
**Overall Recommendation:** 4
**Confidence:** 3

**Summary:**

In this paper, the authors propose an algorithm for AI-Human collaboration. Different from existing work that usually fixes the errors themselves, the proposed method fixes the knowledge gap inside the human. To achieve this, the algorithm conducts knowledge gap localization as well as latent knowledge editing and decoding to train the model. Results show that the proposed algorithm surpasses the GTP and Llama baselines by a large margin.

**Compliance With Llm Reviewing Policy:**

Affirmed.

**Final Justification:**

See Acknowledgement

**Key Questions For Authors:**

See above.

**Limitations:**

yes

**Strengths And Weaknesses:**

Strengths:
1. The idea is intuitive and effective. Correcting the model performance from the root or the knowledge of the human is a good way to eliminate errors for future errors.
2. The experiment results, together with the ablation and user study, prove the effectiveness of the proposed algorithm which is better than the GPT baselines.

Weaknesses:
1. The research scope is unclear. It is not easy to understand the problem that the paper is working on. Is it designed for human-ai collaboration, AI navigation, or assistance? Is the algorithm designed for physical agents, game agents, or for the broader domain? Furthermore, it is also unclear the importance of the scope: is the AI correction important? Why and when do you need this? Are there any established tasks, rather than games, that this can be used for?

2. The experiment setting is over-simplified. First, there are only three types of the pertabations (items, behaviors, workarounds), all with rules.  Can it cover the real-world student errors? How are these samples generated, and how can the quality be controlled? I think some human annotations or quality control can better demonstrate that the error types are sufficient. Second, the metrics are towards the target of the training and the samples, meaning they might be biased towards what is trained. Adding more existing metrics or some other aspects can make it more general.

3. Lacks more baselines. Only GPT and some random baselines are compared; it ignores some specific experts, such as those with training or similar optimization. Since the proposed method works on knowledge correction, some non-knowledge methods can be used as baselines.

4. Some results need an in-depth explanation. In Table 1, the strongest baselines usually only obtain 30% lower score. What is the problem of exisiting methods? Some analysis can better demonstrate how the proposed work solves the problem. In Table 2, SESEI performs lower than the ablations from the Breakfast dataset. The reasons can be a good analysis to increase the robustness of each component of compelx system.

---

> ### Author Rebuttal · Authors · 2026-03-31
>
> We sincerely thank the reviewer for their constructive comments.
>
> ### W1 Scope & Significance
> SENSEI is a general framework for human-knowledge-aware AI assistance, not limited to navigation, games, or a specific embodiment. It applies to domains where agents behave according to their underlying knowledge, and where expert behavior provides a target for alignment. The importance of this scope is that correcting actions alone is often insufficient. Existing approaches often “fix the move,” but do not address the root misconception that caused the error. SENSEI instead aims to “fix the mind,” so that the user updates their mental model and can generalize to future situations. This is why AI correction is important: it is needed when the goal is not just task completion, but robust learning and skill transfer. We evaluate SENSEI on mobile robotics and cooking tasks as representative testbeds, but the framework is broadly applicable to established non-game settings such as industrial training, adaptive user interfaces, and human-robot teaming, wherever expert and novice knowledge-behavior data are available.
>
> ### W2.1 Real-World Complexity
> To clarify, our environments are established benchmarks: Overcooked is a standard benchmark in multi-agent coordination [1, 2, 3, 4], and Rover, adapted from the International Planning Competition [5], is a standard for long-horizon planning [6, 7]. Please find details on knowledge perturbations and complexity of our experimental setting in our response to Reviewer BLzX (W1.1).
>
> ### W2.2 Evaluation Metrics
> Our metrics (recall, precision, F1 score, false positive rate) are universally established in classification, fault diagnosis, and information retrieval [8, 9, 10]. As SENSEI’s Localization Module performs multi-label classification over the knowledge space, these metrics rigorously evaluate the model’s ability to diagnose student misconceptions with minimal false alarms. Beyond knowledge alignment, our user study (Sec. 5.7) measures how well SENSEI aligns real human behaviors to the expert’s, which we believe addresses the reviewer’s suggestion for “other aspects” of evaluation. We welcome suggestions for additional metrics.
>
> ### W3 Comparison to Other Methods
> We agree that non-knowledge-based methods (e.g., ones in Sec. 2) suit immediate behavior alignment. However, our formulation strictly targets knowledge alignment (Eq. 3). As noted in Sec. 5.3, SENSEI is the first framework designed for interpretable knowledge diagnosis. Non-knowledge baselines trivially fail this objective as they do not output interpretable student knowledge. Furthermore, we do include specifically trained, optimization-based baselines. The Monolithic End-to-End baseline (Sec. 5.3) is fully finetuned to directly map behaviors to corrected PDDL blocks. Our Random, Random-H, and Oracle baselines utilize a Knowledge Edit Module specifically trained for this task, isolating editing performance. Finally, we select SoTA LLMs as the strongest off-the-shelf alternative for general purpose reasoning and text generation required for our objective: human-interpretable misconception diagnosis. Please let us know if there are other specific baselines we should consider.
>
> ### W4.1 Baseline Result Discussion
> End-to-end and LLM baselines suffer from low localization recall and edit accuracy (Table 1), bottlenecking system-level performance. As confirmed in Appendix F, LLM failures stem from poor causal reasoning, not just PDDL syntax generation. When asked to explain the student’s misconception, LLMs often describe behavioral divergence (e.g., “Student used bowl2 instead of bowl1”) rather than the underlying cause (student does not know bowl2 is not microwave safe). These results suggest that these baselines struggle to draw causal connections between student mental model and behavior, failing to localize and correct the error source. SENSEI resolves this by decoupling “where to edit” and “how to edit,” and by grounding behavior to specific symbolic knowledge components. The low Localization Precision of the Random baselines demonstrates how localization is a non-trivial task, further highlighting effectiveness of SENSEI’s localization module on overall performance. We thank the reviewer for suggesting improving our analysis of baseline performance, and will include this discussion in Sec. 5.5 and appendix of the final version.
>
> ### W4.2 Ablation Result Discussion
> Please refer to our response to Reviewer BLzX (W1.3).
>
> [1] Carroll et al., arXiv:1910.05789
> [2] Fontaine et al., arXiv:2106.10853
> [3] Hong et al., arXiv:2303.02265
> [4] Wang et al., arXiv:2310.05208
> [5] Long et al., arXiv:1106.5998
> [6] Richter et al., arXiv:1401.3839
> [7] Corrêa et al., arXiv:2503.18809
> [8] Sokolova et al., A Systematic Analysis of Performance Measures for Classification Tasks
> [9] Manning et al., Introduction to Information Retrieval
> [10] Fawcett. An Introduction to ROC Analysis

---

> > ### Author Rebuttal · Reviewer_VPdY · 2026-04-01
> >
> > Thanks for the clarifications. I decide to raise the score to 4.

---

### Decision · Program_Chairs · 2026-04-30

**Decision:**

Accept (regular)

**Comment:**

This paper proposes a framework that infers and repairs users' underlying knowledge gaps from interaction behavior, to address the problem of long-horizon human-AI collaboration where action-level feedback alone is insufficient for correcting recurring mistakes.

Reviewers are overall positive about this paper and acknowledge that the perspective of moving AI assistance from action correction to cognitive diagnosis is novel and well-motivated [Reviewer hRsA]; the framework demonstrates strong zero-shot compositional generalization and is supported by thorough experiments including a user study [Reviewer 3X1V]; and the paper is well-written with an interpretable architecture and clear visualizations [Reviewer BLzX].

Following the rebuttal, all four reviewers confirmed their concerns were fully addressed, with three reviewers raising their scores. There are no major remaining concerns.Therefore, the AC recommends accept for this paper.